# LESS IS MORE: UNDERTRAINING EXPERTS IMPROVES MODEL UPCYCLING

## ABSTRACT

Modern deep learning is increasingly characterized by the use of open-weight foundation models that can be fine-tuned on specialized datasets. This has led to a proliferation of expert models and adapters, often shared via platforms like HuggingFace and AdapterHub. To leverage these resources, numerous model up-cycling methods have emerged, enabling the reuse of fine-tuned models in multi-task systems. A natural pipeline has thus formed to harness the benefits of transfer learning and amortize sunk training costs: models are pre-trained on general data, fine-tuned on specific tasks, and then upcycled into more general-purpose systems. A prevailing assumption is that improvements at one stage of this pipeline propagate downstream, leading to gains at subsequent steps. In this work, we challenge that assumption by examining how expert fine-tuning affects model upcycling. We show that long fine-tuning of experts that optimizes for their individual performance leads to degraded merging performance, both for fully fine-tuned and LoRA-adapted models, and to worse downstream results when LoRA adapters are upcycled into MoE layers. We trace this degradation to the memorization of a small set of difficult examples that dominate late fine-tuning steps. This causes negative parameter interference and encodes knowledge that is forgotten during merging. Finally, we demonstrate that task-dependent aggressive early stopping strategies can significantly improve upcycling performance.

## 1 INTRODUCTION

The rise of open-weight foundation models, such as CLIP (Radford et al., 2021; Ilharco et al., 2021), T5 (Raffel et al., 2020) and the more recent Gemma (Team, 2025), Llama (Grattafiori et al., 2024) and DeepSeek (DeepSeek-AI, 2024), has caused a paradigm shift in the field of machine learning. Instead of training a model from scratch as was previously the norm, it is now increasingly common for practitioners and researchers alike to start with a pre-trained foundation model and then fine-tune it on a task of interest (Stanford-CRFM, 2021). This approach leverages the benefits of transfer-learning, leading to performance and robustness gains. The proposal of multiple parameter-efficient fine-tuning (PEFT) methods (Hu et al., 2022; Liu et al., 2022), which reduce the computational costs of fine-tuning and limit catastrophic forgetting by only updating a subset of the model parameters, further enables this approach. This has lead to a proliferation of different versions of these foundation models and of PEFT adapters, fine-tuned on a variety of downstream tasks, which are openly accessible on public model repositories such as Hugging Face (Wolf et al., 2019) and Adapter Hub (Pfeiffer et al., 2020).

Model *upcycling*, the practice of reusing existing models to create new, more capable deep learning systems (Zhang et al., 2024; He et al., 2024), capitalizes on this proliferation of fine-tuned models and adapters. Two upcycling strategies stand out: *model merging*, and *model MoErging*. Model merging methods combine multiple fine-tuned versions of the same foundational model into one, preserving the size and therefore the computational and memory requirements of the original pre-trained model while infusing it with multiple new capabilities (Matena & Raffel, 2022; Jin et al., 2023; Ilharco et al., 2023; Yadav et al., 2023; Yu et al., 2024; Davari & Belilovsky, 2024). The advent of model merging techniques and open-source libraries for merging (Kandpal et al., 2023; Goddard et al., 2024) has had an important impact on the deep learning community, providing a simple, training-free way to create better models from already existing checkpoints and adapters.

In the past year, many of the top performing models on HuggingFace's Open LLM Leaderboard (Beeching et al., 2023) have resulted from the merging of fine-tuned checkpoints (Yu et al., 2024).

Model MoErging (Yadav et al., 2024) similarly combines multiple adapted experts, but instead of fusing the parameters directly, MoErging approaches such as Ostapenko et al. (2024); Muqeeth et al. (2024) combine adapters into modular, mixture-of-experts (MoE) type layers (Shazeer et al., 2017) expanding the model's size and capabilities. A routing mechanism determines which input, or part of the input, gets processed by which expert modules. For this upcycling strategy further training is often required to let the router and expert adapters learn how to interact with one another.

A natural pipeline has therefore emerged to leverage the benefits of transfer-learning and amortize past sunk training costs: large models are *pre-trained* in an unsupervised fashion on large amounts of general, unlabeled data; these foundational models are then *fine-tuned*, potentially using PEFT techniques, on specialized datasets or tasks; finally these fine-tuned expert checkpoints or adapters are *upcycled* and combined to create more capable, often multi-task models.

A common assumption is that *increased performance at one stage of this pipeline will propagate downstream*. In other words, a stronger pre-trained model should yield a stronger fine-tuned model, and similarly, stronger fine-tuned experts should produce a stronger merged / MoErged model. We challenge this assumption in this work by studying the following questions: *How does expert training affect upcycling?* and *Do all capabilities and knowledge transfer equally well?*

We find that long fine-tuning that optimizes for expert performance can substantially hurt model upcycling, a phenomenon to which we refer as "overtraining" in the context of this paper. While overtrained experts might be better on their respective fine-tuning tasks, they lead to worse performance when merged or when used as initializations for model MoErging. We validate this phenomenon across diverse settings, including merging fully fine-tuned and PEFT models, performing MoErging with LoRA adapters, in both vision and language domains and across different model families and sizes. We use tools from the data difficulty literature to link prolonged training to the memorization of hard examples. This memorization causes negative parameter interference, leading to hard examples being overwhelmingly forgotten during merging, while easy examples remain correctly classified While some recent work has hinted that undertraining experts can benefit merging performance (Pari et al., 2024; Zhou et al., 2025), our work provides a systematic analysis of this phenomenon, and demonstrates how simple early stopping strategies can significantly improve the efficacy of existing merging and MoErging techniques. Our research introduces a critical new dimension to model upcycling, showing how careful expert training, and targeted checkpoint release can unlock improved performance.

Concretely, our contributions are the following:

- We show that overtraining full fine-tuned (FFT) models produces sub-optimal merges (Section 3.1), and that the negative impact is even stronger when using LoRA adapters for parameter-efficient fine-tuning (Section 3.2);
- We explain this phenomenon through the lens of data difficulty in Section 4, showing that later training steps are dominated by the memorization of a small fraction of difficult examples, which are predominantly forgotten during merging due to negative parameter interference.
- We show that for model MoErging, overtraining the constituent experts leads to lower final accuracy after further multi-task training of the modular model (Section 3.3).
- We show that task-dependent expert training duration can further improve upcycling performance. We propose early stopping as a general principle to encourage expert undertraining. Our early stopping strategies effectively adapt training duration per task and can recover optimal upcycling accuracy (Section 5).

## 2 PRELIMINARIES AND METHODOLOGY

### 2.1 MODEL MERGING

Model merging has recently gained a lot of popularity as a means to combine the abilities of multiple fine-tuned versions of the same pre-trained model into one, preserving the model architecture and size (Yang et al., 2024). Formally, a model merging method, $Merge$, takes the parameters $\theta_0$ of the pre-trained foundation model, and parameters $\{\theta_t\}_{t \in \mathcal{T}}$ of the multiple *experts*, which are fine-tuned models on each task $t$ from a set $\mathcal{T}$, and outputs the parameters of the merged model $\bar{\theta} = Merge(\theta_0, \{\theta_t\}_{t \in \mathcal{T}})$. A simple example of this combination step is averaging the different fine-

tuned models' parameters:

$$\bar{\theta} = \frac{1}{|\mathcal{T}|} \sum_{t \in \mathcal{T}} \theta_t. \tag{1}$$

Merging methods are generally motivated by the observation that fine-tuned models exhibit *linear mode connectivity*: their loss minima are connected by low-loss linear paths in parameter space (Frankle et al., 2020; Sharma et al., 2024). This property typically emerges because fine-tuned models share substantial portions of their training trajectories (Frankle et al., 2020; Neyshabur et al., 2020), and is therefore a key reason merging is expected to be feasible. Nonetheless, a common challenge in model merging is the observed performance degradation of the merged model $\bar{\theta}$ on individual tasks $t \in \mathcal{T}$, relative to the original fine-tuned model $\theta_t$. This phenomenon has been coined "interference", and a plethora of merging methods have been proposed to reduce interference when merging models and to preserve as much of the accuracy of the expert models as possible (Matena & Raffel, 2022; Jin et al., 2023; Yadav et al., 2023; Yu et al., 2024; Deep et al., 2024; Davari & Belilovsky, 2024). These methods have mainly focused on modifying the experts parameters $\{\theta_t\}_{t \in \mathcal{T}}$ or the respective *task vectors* $\{\tau_t\}_{t \in \mathcal{T}}$, where $\tau_t = \theta_t - \theta_0$, and / or changing the combination step. We consider 4 popular merging methods:

- **Average** simply averages the parameters of all fine-tuned models following Equation (1);
- **Task Arithmetic (TA)** (Ilharco et al., 2023) scales the sum of the task vectors by a tuned scalar $\lambda$, and adds it to the pre-trained model parameters, returning $\theta_0 + \lambda \sum_{t \in \mathcal{T}} \tau_t$;
- **TIES** (Yadav et al., 2023) prunes low magnitude parameters from each task vector, then only averages the remaining parameters if they have the same sign as the weighted majority;
- **DARE** (Yu et al., 2024) randomly prunes a fraction of each task vector parameters; the remaining parameters are rescaled based on the pruning fraction, and are combined as in TA.

We review other popular methods in Appendix A and detail our hyperparameter tuning procedure for merging in Appendix B. Prior works have primarily focused on deriving new techniques to reduce interference while assuming fixed, standard fine-tuning protocols. The role of the fine-tuning procedure itself, particularly its duration, has received little attention, with some exceptions discussed in Section 6. Our work explicitly studies how expert training time affects mergeability.

## 2.2 MODEL MOERGING

Another popular class of upcycling strategies besides model merging are model MoErging techniques. MoErging methods aggregate multiple fine-tuned experts with the use of modular architectures, such as mixture-of-experts (MoE) layers (Shazeer et al., 2017), to build stronger deep learning systems. The large design space of these methods, paired with their effectiveness has led to the rapid development of many new methods in the recent past, with varying expert, router and application design choices (Yadav et al., 2024; Huang et al., 2024; Ostapenko et al., 2024; Muqeeth et al., 2024). A key feature of MoErging approaches is modularity; multiple experts are considered simultaneously and a routing mechanism decides which input, or part of an input, is processed by which expert.

In this work we consider per-token and per-layer routing, following recent works which suggest this leads to better performance relative to other possible configurations (Ostapenko et al., 2024; Muqeeth et al., 2024). Concretely, let $\mathbf{W} \in \mathbb{R}^{d_{\text{out}} \times d_{\text{in}}}, b \in \mathbb{R}^{d_{\text{out}}}$ denote the weight matrix and bias of a pre-trained linear layer, whose original output is $\mathbf{W}x + b$. We assume the availability of a fine-tuned expert module $E_t(\cdot)$ for each target task $t \in \mathcal{T}$ and we replace the original linear layer with a MoE layer. A router $\pi$ parameterized by matrix $R \in \mathbb{R}^{|\mathcal{T}| \times d_{\text{in}}}$ computes routing logits $Rx$ and applies softmax $\sigma(\cdot)$ to obtain the routing probabilities. The outputs of the experts with top $k$ highest probabilities are then computed and weight-averaged. The resulting MoE layer output is:

$$y = \mathbf{W}x + b + \frac{\sum_{t \in I_k(x)} \pi(x)_t E_t(x)}{\sum_{t \in I_k(x)} \pi(x)_t}, \tag{2}$$

where $I_k(x) = \{t \mid \pi(x)_t \in \text{top k elements of } \pi(x)\}$. We use $k = 2$ for our experiments.

We consider the "multi-task" setting where we assume access to all the datasets the experts were trained on. After updating every linear layer of the pre-trained model with available adapters, we continue training the MoE-fied model on the multi-task mixture of data by freezing the original model parameters and only updating the router and the expert modules. Our MoErging setup is

closely related to Ostapenko et al. (2024), which combines LoRA experts into MoE layers and initializes routers using Arrow, but we additionally assume access to the training data and continue multi-task training. Notably, while prior work spans many routing and architectural designs, none study how expert overtraining affects downstream MoErging performance.

### 2.3 LOW-RANK ADAPTATION

Modern foundation models have tens, if not hundreds, of billions of parameters, making full fine-tuning impractical on typical hardware (Grattafiori et al., 2024; DeepSeek-AI, 2024; Team, 2025). Parameter-Efficient Fine-Tuning (PEFT) updates only a small subset of the parameters to ease the computational burden and curb catastrophic forgetting (Hu et al., 2022; Liu et al., 2022). Low-Rank Adaptation (LoRA) (Hu et al., 2022), has emerged as one of the most popular PEFT methods due to its simplicity and effectiveness. LoRA inserts two low-rank matrices $\mathbf{A}$ and $\mathbf{B}$ into selected linear layers of a model. If the input and output dimension at that layer are $n_{in}$ and $n_{out}$, LoRA uses a rank $r \ll \min(n_{in}, n_{out})$ to define matrices $\mathbf{A} \in \mathbb{R}^{r \times n_{in}}$ and $\mathbf{B} \in \mathbb{R}^{n_{out} \times r}$. The output of that layer then becomes $(\mathbf{W}\mathbf{x} + \mathbf{b}) + \frac{\alpha}{r}\mathbf{B}\mathbf{A}\mathbf{x}$ where $\alpha$ is a scaling hyperparameter. During fine-tuning, the original model is frozen and only the LoRA's $\mathbf{A}, \mathbf{B}$ matrices are updated.

**Merging LoRA adapters**  At each layer, the weight update induced by LoRA is exactly $\Delta W = W_{\text{fine-tuned}} - W_{\text{pre-trained}} = \frac{\alpha}{r}\mathbf{B}\mathbf{A}$. Consequently, standard merging techniques can be directly applied to LoRA-adapted models if the updates $\frac{\alpha}{r}\mathbf{B}\mathbf{A}$ are added to the pre-trained weights or if they are directly used to compute the task vectors. Merging the LoRA $\mathbf{A}$ and $\mathbf{B}$ matrices separately is not recommended since this can lead to mismatched representation spaces resulting in poor performance (Stoica et al., 2025). Nevertheless, recent work has observed that merging LoRA-adapted models is harder than merging FFT models (Tang et al., 2024a; Stoica et al., 2025), often leading to significant performance degradation.

**Model MoErging with LoRA adapters**  Using LoRA adapters for model MoErging is straightforward, with each adapter being used to define one expert module in the MoE layer. Let $\mathbf{A}_t$ and $\mathbf{B}_t$ denote the LoRA low-rank matrices obtained from fine-tuning on task $t$, then we can define the expert modules in Equation (2) as $E_t(x) = \frac{\alpha}{r}\mathbf{B}_t\mathbf{A}_t x$ for each task of interest $t \in \mathcal{T}$.

### 2.4 DATA DIFFICULTY

In this work, we use data difficulty scores to identify which knowledge is transferred during upcycling and to relate merging performance to training dynamics and memorization. Specifically, we use the EL2N score introduced by Paul et al. (2021) which measures the norm of the error vector, i.e. the predicted class probabilities minus the one-hot label encoding. For a training example $x$ with one-hot label $y$, the EL2N score is defined as $\mathbb{E}\|p(\theta, x) - y\|_2$, where $p(\theta, x)$ denotes the model's predicted class probabilities for $x$ under parameters $\theta$.

Prior work has examined how individual data points influence neural network training dynamics and properties such as generalization, memorization, and privacy, leading to the development of various data difficulty scores (Kwok et al., 2024). These scores aim to quantify an intrinsic characteristic of the data, namely *data difficulty*, which captures how easy or hard individual examples are to learn. Easy examples typically exhibit common, easily learnable features, whereas hard examples often possess idiosyncratic structure or even noisy labels. Such scores have been successfully used for data pruning, with past work showing that large fractions of easy examples can be removed with little effect on performance, while pruning a small fraction of the hardest examples can improve generalization by eliminating outliers with uncommon features (Toneva et al., 2019) or mislabeled data (Paul et al., 2021). Moreover, Sorscher et al. (2022) showed that appropriate data pruning can yield better-than-power-law error scaling with dataset size.

A natural relationship exists between data difficulty and deep learning generalization and memorization. For instance, Sorscher et al. (2022) found a 0.78 Spearman rank correlation between EL2N scores (Paul et al., 2021) and the memorization score presented by Feldman & Zhang (2020). These observations suggest that correctly classifying difficult examples often requires memorization, and that a certain degree of memorization is therefore necessary for achieving near-optimal generalization. This relationship between memorization and generalization has been further substantiated with theoretical results in simpler settings (Attias et al., 2024; Feldman, 2020).

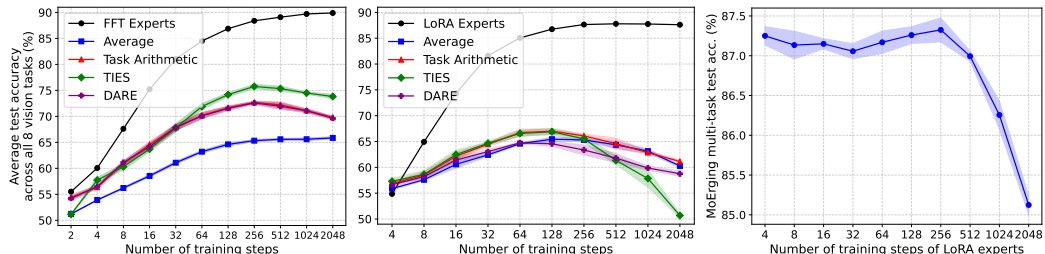

Figure 1: Average test accuracy across the 8 vision classification tasks for merged and MoErged ViT-B-32 experts. **Left:** merging fully fine-tuned experts, we plot the average accuracy of the expert models evaluated on their respective tasks as well as merging accuracies for multiple methods; **Center:** merging LoRA-adapted experts; **Right:** final multi-task accuracy of MoE-fied models vs. LoRA training steps used for initialization. Shaded regions show mean±std over 3 random seeds.

## 2.5 MODELS AND DATASETS

**Vision domain**    We evaluate merging performance in a standard vision benchmark setting using the official codebase from Ilharco et al. (2023): a CLIP (Radford et al., 2021) pre-trained ViT-B-32 model (Dosovitskiy et al., 2021) is fine-tuned on 8 image classification tasks: Cars (Krause et al., 2013), DTD (Cimpoi et al., 2014), EuroSAT (Helber et al., 2019), GTSRB (Stallkamp et al., 2012), MNIST (Deng, 2012), RESISC45 (Cheng et al., 2017), SUN397 (Xiao et al., 2010) and SVHN (Netzer et al., 2011). The fine-tuning is done with a batch size of 128, the AdamW optimizer (Loshchilov & Hutter, 2019; Paszke et al., 2019) and a learning rate of 1e-5. We use a learning-rate scheduler with linear warm-up for the first 10% of training, followed by cosine annealing. When evaluating merged models, we use the corresponding frozen classification head for each task.

**Language domain**    For our natural language processing (NLP) experiments, we adopt the setting of the TIES paper (Yadav et al., 2023) and use their released code. We use pre-trained T5-Base models (Raffel et al., 2020) which we fine-tune on 7 tasks: QASC (Khot et al., 2020), WikiQA (Yang et al., 2015) and QuaRTz (Tafjord et al., 2019) for question answering; PAWS (Zhang et al., 2019) for paraphrase identification; Story Cloze (Sharma et al., 2018) for sentence completion and Winogrande (Sakaguchi et al., 2020) and WSC (Levesque et al., 2012) for coreference resolution. We use the AdamW (Loshchilov & Hutter, 2019) optimizer with a batch size of 256, a constant lr of 0.0001 and no weight decay. bfloat16 mixed precision training is used to reduce GPU utilization.

**Evaluation**    For all our experiments we report the raw, un-normalized test accuracy averaged across the multiple considered tasks. We chose not to use the popular *normalized accuracy* metric because the set of experts being merged here differs across experiments, which also changes the normalization factor and makes comparisons inconsistent. A more detailed justification is provided in Appendix C. Our experiments are ran using the PyTorch (Paszke et al., 2019) and HuggingFace (Wolf et al., 2019) open source machine learning frameworks on an Nvidia Quadro RTX 8000 GPU with 48GB of memory.

## 3 LONGER FINE-TUNING HURTS MODEL UPCYCLING

In this section, we present results challenging the common assumption that better fine-tuned models lead to better upcycling results. We show that overtrained experts lead to worse merged models for both FFT and LoRA, as well as lower accuracy when used to initialize MoErging methods.

### 3.1 MERGING FULLY FINE-TUNED MODELS

While a multitude of model merging methods have been proposed, the influence of the fine-tuning procedure itself on merging remains understudied. Most prior works have used similar fine-tuning protocols, typically training for a fixed 2000 steps in the vision setting described in Section 2.5. Instead of proposing yet another model merging method, we take a look at how the number of training iterations affects merging. We fine-tune our vision and NLP models for varying number of training steps $s \in \{2, 4, 8, 16, 32, 64, 128, 256, 512, 1024, 2048\}$ on every considered dataset. Each merge combines either 8 vision or 7 NLP experts (one per task) all trained for the same duration.

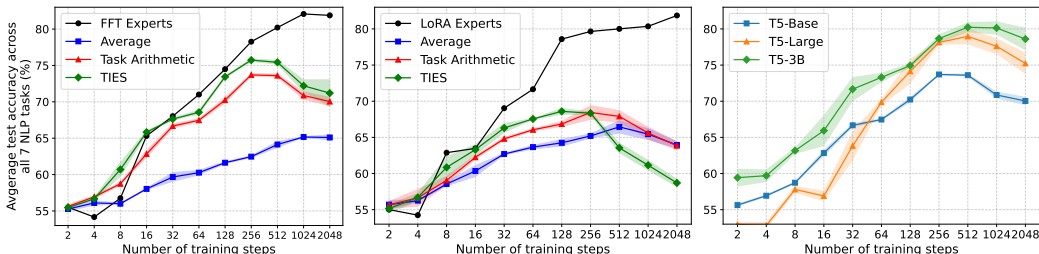

Figure 2: Average test accuracy across all 7 NLP tasks for fully fine-tuned (**left**) and LoRA-adapted (**center**) T5-Base models. We plot the average accuracy of the expert models evaluated on their respective tasks as well as merging accuracies for multiple methods. **Right:** Task Arithmetic merging accuracy for different T5 model sizes. Shaded regions show mean±std over 3 random seeds.

**Undertrained experts result in better merging**    Figure 1 (left) shows that, except for Average, all methods achieve better merging performance when the ViT experts are trained for just 256 training steps, only ∼1/8 of the commonly used 2000. TA, TIES, and DARE yield models with ∼3% higher accuracy at 256 steps compared to 2048, a gain comparable to the 3.4% gap between TA and the more sophisticated TIES at 2048 steps. The same conclusions hold in the NLP setting (Figure 2 left), with both TA and TIES peaking around 256–512 training steps. Further training leads to a drop in merging performance of over 3% for both merging methods. Notably, merging undertrained experts with TA outperforms merging experts trained for longer with TIES. Average is the only method that seems to benefit from training the experts longer, but it consistently underperforms overall. Moreover, TA, TIES, and DARE show similar trends across training durations, suggesting that training length itself, rather than the merging method, plays a key role in merging performance.

**Better experts do not necessarily lead to better merging**    The black lines in the left and central panels of Figures 1 and 2 show the average accuracy of the expert models on their respective fine-tuning tasks. In both the vision and NLP settings, we observe that higher expert accuracy does not necessarily translate into better merging performance. In the vision setting, expert models trained for 256 steps achieve an average accuracy of 88.4%, which is 1.6% lower than at 2048 steps (90.0%). Nevertheless, merging after 256 steps yields models with approximately 3% higher accuracy than merging after 2048 steps. The discrepancy is even more pronounced in the NLP setting. Expert accuracy improves from 78.2% at 256 steps to 82.4% at 1024 steps, a 4% gain, yet the merging accuracy of TA and TIES drops by around 3% over the same interval. We provide a per-task breakdown of these results in Appendix D and further experiments with ViT-L-14 and BERT models in Appendix E. The same "overtraining degrades upcycling" phenomenon can be generally observed.

**Effect of model scale**    In the right panel of Figure 2, we compare Task Arithmetic merging accuracy across different model sizes in the T5 family: T5-Base (220M parameters), T5-Large (770M), and T5-3B (3B). We observe that the same trend persists across scales: upcycling performance peaks at an intermediate number of training steps before degrading with longer fine-tuning. Additional merging results are provided in Appendix F.

### 3.2  MERGING LoRA ADAPTERS

We now extend our previous results to the highly relevant setting of merging LoRA adapters. We find that long training of LoRA experts hurts merging performance even more than in the FFT case. We add LoRA adapters at every linear layer of the original ViT-B-32 and T5-Base models. We use LoRA rank $r = 8$, scaling parameter $\alpha = 32$ and learning rates 1e-4 and 5e-4 for the ViT and T5 models respectively. We train the LoRAs for different number of steps $s$ to evaluate the impact of training duration on accuracy and mergeability. The parameters of the base model are kept frozen.

**Overtraining severely impairs LoRA merging**    The center panels of Figures 1 and 2 show expert and merging accuracies for our vision and NLP LoRA models, respectively. For the ViT models, merging performance peaks at 128 training steps (64 for DARE), with accuracies ranging from 65–67% across all methods. Although further training improves expert accuracy by about 1%, it significantly degrades merging performance, with accuracy drops of 5–6% for Average, TA, and DARE, and nearly 17% for TIES. In the NLP setting, different methods reach peak merging performance at different training durations: 512 steps for Average (66.5%), 256 for TA (68.5%), and 128

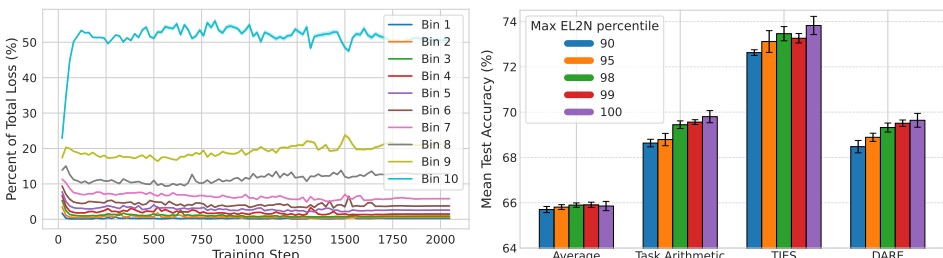

Figure 3: **Left:** Percentage of total loss for examples in different data difficulty bins. Bin 1 represents 10% easiest examples (lowest EL2N scores), bin 10 represents 10% hardes examples (highest EL2N scores). Mean across all 8 vision datasets shown. **Right:** Merging accuracy for experts trained without the hardest examples. Experts are trained on data with EL2N scores from percentile 0 to varying max percentiles in $\{90, 95, 98, 99, 100\}$.

for TIES (68.6%). Expert models, however, continue to improve, reaching an average accuracy of 81.9% at 2048 steps. Despite this, merging at 2048 steps harms performance, with drops of 2.5%, 4.6%, and 9.9% for Average, TA, and TIES, respectively. In Appendix G, we examine the impact of LoRA rank and show that higher ranks lead to smaller performance degradations when merging.

### 3.3 MODEL MOERGING WITH LORA EXPERTS

We next analyze how the performance of MoE-fied models, initialized with LoRA experts, is affected by the training time of these experts. We use the LoRA adapters obtained in Section 3.2 with different number of training steps to initialize our MoE experts, one LoRA for each task. The routing mechanism is initialized using Arrow (Ostapenko et al., 2024), where the weight vector associated with each expert is the first right-singular vector of the $BA$ matrix multiplication. These vectors are assumed to determine the direction of most variance induced by expert $E_t$ for $t \in \mathcal{T}$ in the space of hidden states induced by data from task $t$.

We create one MoE-fied model for each number of steps $s$, i.e. for each different model we initialize the MoE layers with the expert LoRAs for each task, all trained for $s$ steps. Once the MoE-fied model has been initialized using the fine-tuned LoRAs, we further train the routing mechanism and the LoRA experts in a multi-task fashion for 4000 steps with a peak learning rate of 1e-5, with the base parameters frozen. We report the final, multi-task, accuracies over the 8 classification tasks in the right panel of Figure 1.

We observe that the MoE-fied models initialized with overtrained LoRA experts reach about 2% lower final multi-task accuracy than the models initialized with experts trained for less. Even expert LoRAs trained for as little as 4 steps on their respective tasks reach a higher final multi-task accuracy than those overtrained. We conclude that overtraining experts can hurt downstream MoErging.

## 4 WHY IS UNDERTRAINING BENEFICIAL FOR MERGING?

In this section we use tools from the data difficulty literature to explain why undertraining is beneficial for model merging. This allows us to make a series of empirical observations linking prolonged training to the memorization of hard examples and to increased parameter interference.

For all the training examples from the 8 considered image classification tasks we compute the EL2N data difficulty scores early in fine-tuning, after only 32 steps, across 10 different seeds (different models than the ones we merge). We note that despite being computed early in training, the EL2N scores aim to estimate an intrinsic characteristic of the data, independent of model training. To facilitate analysis, we group the training examples into 10 bins according to their EL2N scores, the 10% of examples with the lowest EL2N scores (the easiest examples) are in bin 1 and so on.

**Observation 1: Later training stages are driven by the memorization of hard examples.** In the left panel of Figure 3 we show the relative loss of the training examples during training. We observe that easy examples, which have more common features, are learned very early in training. **The remaining of training is driven largely by the loss of the difficult examples**, with the top 10% of hardest examples accounting for over 50% of the total loss after the first 100 steps. In Appendix H

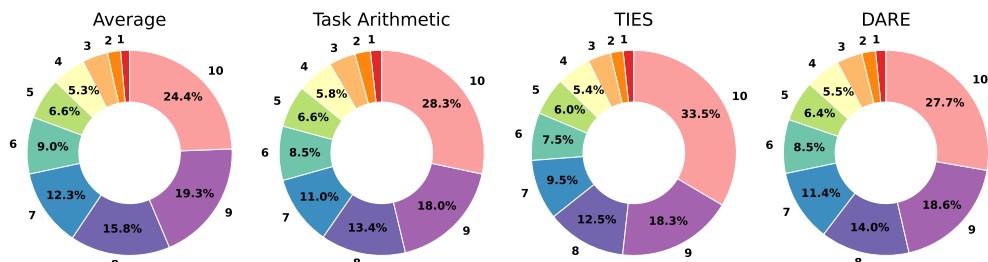

Figure 4: Proportion of forgotten examples in each data difficulty bin for three different model merging methods. Bin 1 represents 10% easiest examples (lowest EL2N scores), bin 10 represents 10% hardest examples (highest EL2N scores). Hard examples are overwhelmingly forgotten when merging with all methods, with the 30% hardest examples representing over 50% of forgotten examples.

we quantify memorization of examples in each data difficulty bin using three metrics: *change in margin* (gap between true class probability and maximum probability among other classes), *change in loss*, and *predictive distribution shift* (L1 norm of the difference between predicted probabilities). Comparing ViT-B-32 models trained for 256 and 2048 steps, all metrics show changes orders of magnitude larger for the hardest examples (bin 10) than for the easiest (bin 1), despite only modest accuracy gains on the test set (1.6% on average) and on the training set (4%). This suggests that **memorization of hard examples occurs during the later stages of training**.

**Observation 2: Later training stages result in idiosyncratic parameter updates causing increases in parameter interference.** As discussed in Section 2.4, hard examples represent outliers with uncommon features or noisy labels. Therefore, it stands to reason that the memorization of such examples would also yield idiosyncratic parameter updates that don't generalize across tasks. In Appendix I we estimate the amount of parameter interference between the task vectors obtained with various training durations using four different metrics: *sign conflict percentage*, *parameter overlap percentage*, *magnitude ratio* and *per-parameter variance*. All of these parameter interference scores increase with training duration, confirming that **longer training yields idiosyncratic parameter updates leading to more parameter interference.** Since parameter interference is a well accepted explanation to the performance degradation observed when merging (Yadav et al., 2023), **this directly explains our main observation that longer training negatively impacts model merging**.

To directly link these parameter updates to the memorization of hard examples, we take a look at which examples are *forgotten* during merging, i.e. training examples correctly classified by the expert models but incorrectly classified after merging. Figure 4 shows that hard examples are disproportionately forgotten during merging, with over 50% of forgotten data points in the top 30% in terms of data difficulty. This confirms that parameter updates from prolonged training are idiosyncratic and driven by memorization of hard examples, and that merging destroys some some of these learned features due to parameter interference.

**Observation 3: Difficult examples are still necessary for good upcycling generalization.** Inspired by work showing that removing difficult examples from training can aid generalization (Toneva et al., 2019; Paul et al., 2021), we investigate how this affects merging performance. We remove the top 1, 2, 5, or 10% most difficult examples from training and report merging results in the right panel of Figure 3. However, we find that the best merging results are achieved when all data is used and that removing difficult examples consistently hurts performance. This expands upon past work showing that some memorization is necessary for close-to-optimal generalization (Feldman, 2020; Feldman & Zhang, 2020; Attias et al., 2024) by demonstrating that **some amount of memorization is also necessary for close-to-optimal merging performance.**

## 5 AGGRESSIVE EARLY STOPPING IMPROVES UPCYCLING RESULTS

Our core finding that overtraining experts harms upcycling performance naturally motivates the use of early stopping to mitigate this effect. We hypothesize that early stopping during fine-tuning will yield stronger upcycling results, as it both shortens training duration and automatically adapts the stopping time for each task. In this section we investigate this hypothesis and find that upcycling can indeed be improved by early stopping, even when expert-level performance is lower. Our goal

Table 1: Experts and merging accuracy (%) for the overtrained, optimal and early stopped ViT-B experts. Mean and standard deviation across 3 random seeds shown.

|  | Experts | Average | Task Arithmetic | TIES | DARE |
|---|---|---|---|---|---|
| FFT 2048 steps | 89.9±0.1 | 65.9±0.2 | 69.8±0.27 | 73.8±0.4 | 69.6±0.3 |
| FFT best (# steps) | - | 65.9±0.2 (2048) | 72.7±0.2 (256) | 75.8±0.4 (256) | 72.6±0.3 (256) |
| FFT early stop | 87.9±0.2 | 64.5±0.1 | 72.6±0.5 | 74.7±0.3 | 72.5±0.5 |
| LoRAs 2048 steps | 87.6±0.1 | 60.3±0.3 | 61.1±0.3 | 50.7±0.4 | 58.8±0.5 |
| LoRAs best (# steps) | - | 65.4±0.4 (128) | 67.0±0.5 (128) | 66.9±0.6 (128) | 64.7±0.1 (64) |
| LoRAs early stop | 87.9±0.1 | 65.6±0.4 | 68.0±0.8 | 67.1±0.6 | 65.3±0.3 |

here is not to establish the optimality of a single strategy, but to propose early stopping as a general principle for mitigating the negative effects of overtraining in upcycling and to introduce practical, automated variants that follows established best practices.

## 5.1 VISION SETTING — LINEAR WARM-UP AND ADAPTIVE DECAY

The learning rate scheduler used in Section 3 for ViT models, a linear warm-up followed by cosine decay, is a standard choice in vision training and recent merging work, where warm-up provides early stability and decay supports smooth convergence. Our early stopping strategy for these models retains this warm-up/decay structure while making it adaptive to the shorter, task-dependent training lengths induced by early stopping. We achieve this by pairing a 50-step linear warm-up phase with a "reduce learning rate on plateau" phase that gradually decreases the learning rate when validation accuracy stagnates. We compute the validation accuracy every 5 training steps and multiply the learning rate by 0.5 when it has not improved for 3 consecutive rounds. Once the learning rate falls below a threshold of 1e-7, training is stopped. We fine-tune FFT and LoRA models on the 8 considered vision tasks using peak learning rates of 1e-5 and 1e-4 respectively. Pseudocode for this LR scheduler and early stopping strategy is provided in Appendix J.

In Table 1 we compare the merging of early stopped experts to two baselines from Section 3: merging "overtrained" models trained for 2048 steps and merging the checkpoints that achieved the highest accuracy among all training durations (same duration across tasks). We see that the models trained using our simple task-dependent early stopping strategy yield merges that are better than those of overtrained models and as good, if not better, than the best merged experts obtained with a common stopping time even though the experts are on worse on their respective tasks. Early stopping seems to work especially well for LoRA adaptation, yielding results on average better than the best ones from Section 3.2.

We also use the early-stopped LoRAs to initialize MoE layers and continue training in a multi-task fashion, as in Section 3.3. As shown in Table 2, the MoErged models initialized with the early stop LoRAs achieve the same accuracy as the best LoRAs across all training steps.

Table 2: Early stopping MoErging results

| Expert initialization | Avg. accuracy |
|---|---|
| 2048 steps LoRAs | 85.1 ± 0.1 |
| Best LoRAs (256 steps) | 87.3 ± 0.2 |
| Early stop LoRAs | 87.3 ± 0.1 |

## 5.2 NLP SETTING — STOPPING WHEN VALIDATION ACCURACY PLATEAUS

Our T5 models from Section 3.1 were trained with a constant learning rate, making the design of the early stopping criterion straightforward: we evaluate accuracy on a held-out validation set every `#steps` batches and stop training whenever the validation accuracy fails to improve for `#wait` consecutive evaluations. The checkpoint with the highest validation accuracy is selected as the expert. We study two variants of this strategy: **v0**, with `#steps`=100 and `#wait`=5, and a more aggressive **v1**, with `#steps`=50 and `#wait`=3. Results are presented in Table 3.

Experts obtained with v0 match the performance of models trained for 1024 or 2048 steps, while v1 yields slightly lower accuracy. Importantly, both strategies achieve these results with far fewer training iterations, on average only 485 steps for v0 and 269 steps for v1. We also note high variance for the number of steps across tasks. Despite the reduced training time, the merging performance improves substantially. Task Arithmetic and TIES both benefit: across seeds, both early stopping strategies produce merges that are roughly 4% more accurate than merges from experts trained for 1024 or 2048 steps. Moreover, the merging results for early stopped models are even superior to the best results from Section 3.1 where the single best stopping point is selected commonly for all tasks. Average merging is the only method which doesn't seem to benefit from early stopping.

Table 3: Merging accuracy (%) and number of training steps for the overtrained, optimal and early stopped T5 experts. Mean and standard deviation across 3 random seeds shown.

|  | # steps | Experts | Average | Task Arithmetic | TIES |
|---|---|---|---|---|---|
| 1024 steps | 1024 | 82.4±0.3 | 65.2±0.1 | 70.9±0.6 | 72.2±1.0 |
| 2048 steps | 2048 | 82.0±0.5 | 65.0±0.4 | 70.1±0.7 | 72.2±1.4 |
| Best (# steps) | – | – | 65.2±0.1 (1024) | 73.7±0.2 (256) | 75.7±0.2 (256) |
| Early stop v0 | 485±350 | 82.3±0.6 | 63.4±0.4 | 75.0±0.3 | 76.9±0.5 |
| Early stop v1 | 269±199 | 81.3±0.4 | 62.8±0.4 | 74.0±0.1 | 77.0±0.3 |

## 6 RELATED WORK

**Special fine-tuning procedures for better merging**   Several recent works study how modifications to fine-tuning can improve merging performance. However, these approaches adjust the fine-tuning procedure itself, for example by updating only selected submodules (Jin et al., 2025) or parameters (Iurada et al., 2025), using sharpness-aware optimization (Lee et al., 2025), or employing linearization-based updates (Tang et al., 2024b; Ortiz-Jimenez et al., 2023).  In contrast, our work analyzes how *standard* fine-tuning protocols, without merging-specific adjustments, affect downstream merging and MoErging performance, providing a complementary perspective on merging behavior under the most commonly used training procedures.

**Expert training time**   Most model merging and MoErging papers do not examine how expert fine-tuning affects downstream upcycling, with two notable exceptions. Zhou et al. (2025) show that the effectiveness of task-vector based approaches is largely driven by first-epoch gradients and propose alternating 1-epoch fine-tuning and merging. While they note that less training can improve accuracy, they only test 1, which can yield either overtrained or undertrained experts depending on dataset size. Pari et al. (2024) observe representational incompatibilities when merging highly specialized experts, but study only two-model merges and propose bypassing merging altogether by using MoErging. To our knowledge, we are the first to systematically link expert training duration to downstream upcycling outcomes, analyze merging through example difficulty, and propose an early-stopping strategy that adapts to dataset heterogeneity. Finally, although TIES Merging (Yadav et al., 2023) uses early stopping, it is only used to avoid expert overfitting and its effect on merging is not studied.

**Overtraining in pre-training**   Analogous to our work, others have studied how scaling pre-training impacts downstream fine-tuning. In a large-scale vision study, Abnar et al. (2022) found that as pre-training accuracy improves, fine-tuning saturates. Recently, Springer et al. (2025) show that over-training LLMs during pre-training can harm fine-tuned in- and out-of-distribution performance.

## 7 CONCLUSION

In this paper, we challenged the assumption that better fine-tuned experts yield better upcycling performance. Across multiple merging methods, model families and sizes, and for both fully fine-tuned and LoRA-adapted models, we found that optimal merging occurs well before full convergence, often when experts are less accurate on their original tasks. For MoErging, continued fine-tuning of LoRA experts even degrades downstream multi-task performance. We attribute this to a shift in training dynamics: as fine-tuning progresses, training becomes dominated by memorization of difficult examples, leading to idiosyncratic parameter updates and negative parameter interference. Finally, we show that simple early stopping strategies mitigate overtraining and can even yield superior upcycling performance.

Our findings have important implications for the sharing of model versions and adapters and the evaluation of upcycling pipelines. **Publish intermediate checkpoints:** Releasing not only final but also intermediate checkpoints is crucial, as the best upcycling point may precede convergence. In practice, even a single intermediate checkpoint, extracted when validation accuracy starts to plateau, is likely sufficient for achieving close-to-optimal upcycling performance, as supported by our early stopping experiments. **Prioritize early-stopped experts:** When training experts in-house, aggressive early stopping can outperform convergence for downstream upcycling. Since upcycling reuses checkpoints and amortizes sunk costs, our findings can help reduce the future computational and environmental footprint of training AI models.

## 8 REPRODUCIBILITY STATEMENT

We have taken several steps to ensure the reproducibility of our work. In Section 2.5 we describe the exact models, datasets, and codebases used, as well as the machine learning frameworks and hardware employed. All frameworks and codebases are open-sourced and publicly available, and our exact codebase is provided as supplementary material. In addition, our main text and the Appendix include all relevant details and a description of our hyperparameter tuning procedures, ensuring that our experiments can be fully reproduced.

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

## A  ADDITIONAL RELATED WORK

The simplest approach, parameter averaging, was shown to lead to better generalization when used on checkpoints from the same training trajectory (Izmailov et al., 2018) and was popularized in federated learning with FedAvg (McMahan et al., 2017). Recently, parameter averaging was also shown to be useful in the context of robust fine-tuning (Wortsman et al., 2022) and to obtain better pre-trained models (Choshen et al., 2022). When merging multiple fine-tuned versions of the same pre-trained model, Fisher-weighted averaging (Matena & Raffel, 2022) and related methods improve upon this simple averaging by adjusting per-parameter contributions (Jin et al., 2023; Tam et al., 2024). Task arithmetic based methods rely on the computation of task vectors, which are then summed, scaled and added back to the pretrained model (Ilharco et al., 2023) to give it multi-task capabilities. Pruning the task vector parameters (Yadav et al., 2023; Davari & Belilovsky, 2024; Yu et al., 2024; Deep et al., 2024) and selectively combining them to reduce negative interference (Yadav et al., 2023) further benefits performance.

Sharma et al. (2024) explores merging of experts that were trained from different or poorly performing pre-trained models.

## B  TUNING MERGING HYPERPARAMETERS

Several merging methods require careful hyperparameter tuning to achieve optimal performance. In particular, Task Arithmetic, TIES, and DARE each apply a scaling factor $\alpha$ to their task-vector sums before adding them to the pretrained weights; TIES and DARE additionally specify a percentage $k$ of weights to retain after pruning. As is standard, we select the best $\alpha$, $k$ values by maximizing merging accuracy on a held-out validation set. All merging accuracies reported in the main text are evaluated on the test set using hyperparameters selected via validation performance. We followed the hyperparameter configurations from the original papers (Ilharco et al., 2023; Yadav et al., 2023; Yu et al., 2024), adjusting them as needed to optimize performance in our experimental settings.

**Crucially, we perform tuning of merging hyperparameters for each individual merging experiment, i.e. the tuning is done for each different number of training steps and each different random seed across all our settings.**

**Vision setting:**  Following (Ilharco et al., 2023), we reserve 10% of the training data for validation and train the ViT models on the remaining 90%. We tune the following hyperparameter values using the validation set:

- **Task Arithmetic:** $\alpha \in \{0.05, 0.1, \ldots, 1\}$
- **TIES:** $\alpha \in \{0.5, 0.6, \ldots, 1.5\}$ and $k \in \{10, 20, 30\}$
- **DARE:** $\alpha \in \{0.05, 0.1, \ldots, 0.55\}$ and $k \in \{10, 20, 30\}$

**NLP setting:**  We adopt the validation splits from (Yadav et al., 2023) and evaluate the following hyperparameter values:

- **Task Arithmetic:** $\alpha \in \{0.1, 0.2, \ldots, 1\}$
- **TIES:** $\alpha \in \{0.8, 0.9, \ldots, 2.1\}$ and $k \in \{10, 20, 30\}$

## C  USING THE RAW, UN-NORMALIZED ACCURACY

The *normalized accuracy* is a very common metric used to compare model merging methods (Ilharco et al., 2023; Yadav et al., 2023). However, because the normalized accuracy depends on both the merged model's performance and that of the experts, it isn't suitable for settings like ours where different sets of experts are used and compared.

The core issue is that normalized accuracy, defined as (merged_accuracy / expert_accuracy), is a relative metric designed to compare different merging methods when the set of experts (the denominator) is fixed. Papers that propose a novel model merging method are justified in using this metric by the fact that they have a fixed set of experts and they are comparing merging methods, therefore

only the numerator changes. In our study, the experts themselves are the primary variable, as their training duration and performance change in each experiment, therefore the denominator changes from one merging experiment to another. This creates paradoxical situations that make the metric misleading for our purposes. For example consider the following scenario:

- **Case 1 (Undertraining):** Experts trained for only a few steps have very low absolute accuracy (e.g., 60%). When merged, they interfere very little since they're all relatively close in parameter space to the zeroshot model, so the merged model also achieves around 60% accuracy. This yields a normalized accuracy near 100%, despite the models being bad at solving the considered tasks.

- **Case 2 (Optimal Training):** Experts trained for longer have high accuracy (e.g., 90%). Merging them results in a high-performing model with 85% absolute accuracy. However, the normalized accuracy is only $85/90 = 94.4\%$ due to negative interference caused by longer training.

Comparing the "useless" 100% from Case 1 with the "useful" 94.4% from Case 2 is meaningless. Absolute, un-normalized accuracy on the other hand allows for a fair and interpretable comparison of the final upcycled model's quality across different expert training durations.

# D PER-TASK BREAKDOWN OF EXPERT AND MERGING ACCURACIES

Here we present per-task accuracy plots for the expert and merged models in both the vision (Figure 5) and NLP settings (Figure 6). While there is some variability to the magnitude of the overtraining effect across datasets and merging methods, the direction of the effect is fairly consistent: for most tasks and merging methods, experts trained for significantly fewer steps yield higher merging accuracy. Importantly, we do not rely on any single dataset or outlier task to support our conclusions: the phenomenon is robust across tasks, domains, and merging methods.

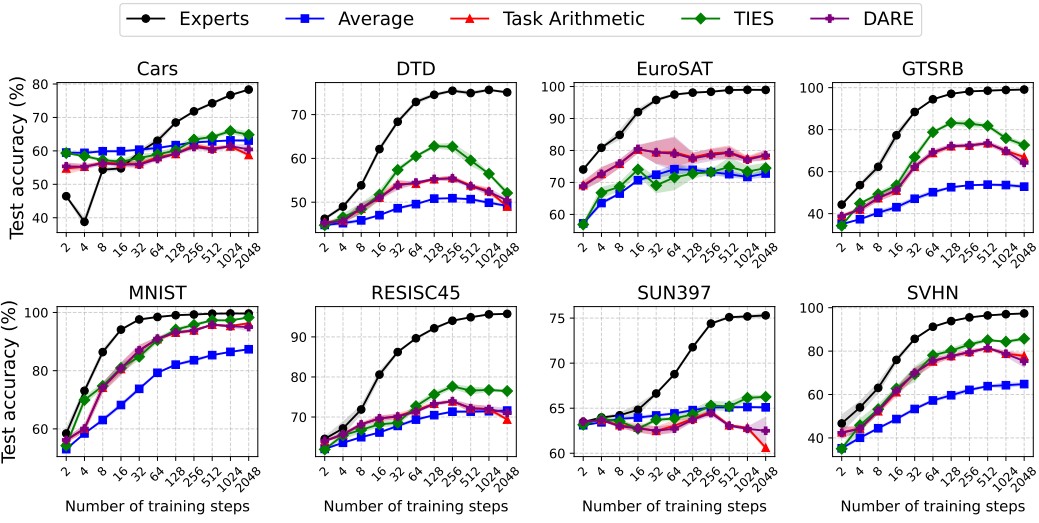

Figure 5: Per-task breakdown of expert and merging accuracies for our ViT-B-32 models trained on 8 image classification tasks. Mean and standard deviation across 3 random seeds shown.

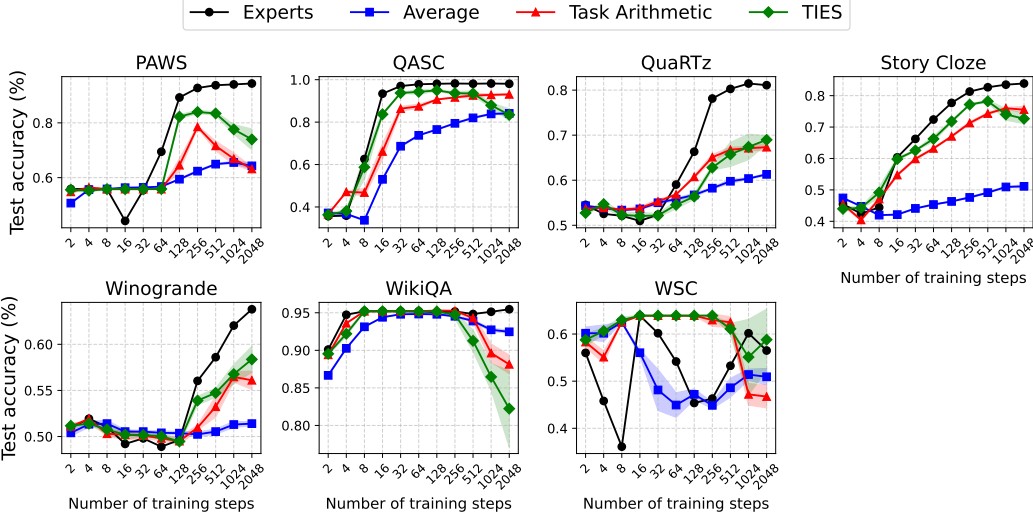

Figure 6: Per-task breakdown of expert and merging accuracies for our T5 models trained on NLP tasks. Mean and standard deviation across 3 random seeds shown.

# E  RESULTS WITH ADDITIONAL MODELS

To further reinforce the generality of our results, we have also run experiments with other model architectures.

## E.1  VISION - ViT-L-14

In the vision setting we have fine-tuned CLIP (Radford et al., 2021) ViT-L-14 (Dosovitskiy et al., 2021) models on the same set of 8 image classification tasks introduced in Section 2. We used the same training hyper-parameters as for the ViT-B-32models. The results are shown in Figure 7. The same phenomenon can be oserved for the larger ViT-L-14 models, while the average expert accuracy keeps increasing throughout training, the average merging accuracy peaks early in training. Task Arithmetic reaches a peak accuracy of 85.6% at 512 steps before decreasing to 84.0% at 2048 steps. TIES reaches a peak accuracy of 87.7% at 512 steps of training before decreasing to 86.2% at 2048 steps. DARE also peaks at 512 steps with an average accuracy of 85.6% before decreasing to 84.0% at 2048 steps. Lastly, Average merging again seems more robust to the negative effects of overtraining on model merging, as in our main results. It continues improving with training reaching a peak accuracy of 79.4% at 2048 steps, only slightly better than the 79.1% achieved at 512 steps. However Average merging yields significantly worse results than all other considered methods, regardless of the number of fine-tuning steps.

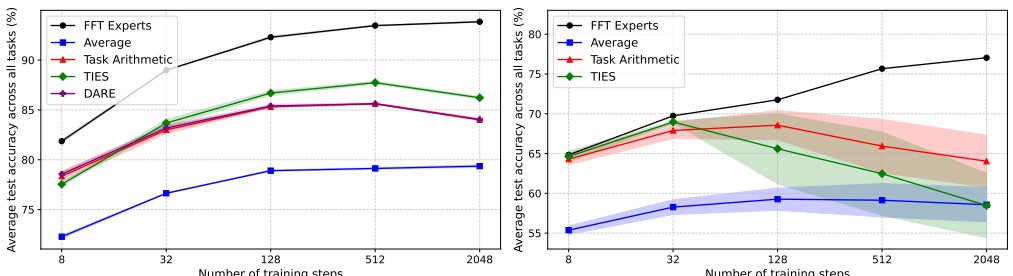

Figure 7: **(left)** Average test accuracy across all 8 vision tasks for fully fine-tuned ViT-L-14 models. **(right)** Average test accuracy across all 7 NLP tasks for fully fine-tuned BERT base models. We plot the average accuracy of the expert models evaluated on their respective tasks as well as merging accuracies for multiple methods. Shaded regions show mean±std over 3 random seeds.

## E.2  NLP - BERT BASE

In the NLP setting we have fine-tuned BERT base (Devlin et al., 2019) language models on the same set of 7 natural language processing tasks introduced in Section 2. For training we used a constant learning rate of 2e-5, the remaining hyper-parameters being the same as for the T5 models. The results are shown in Figure 7. The same phenomenon can be oserved for the BERT models, while the average expert accuracy keeps increasing throughout training, the average merging accuracy peaks early in training. TIES reaches a peak accuracy of 69.0% at only 32 steps of training before decreasing to 58.5% at 2048 steps. Task Arithmetic reaches a peak accuracy of 68.6% at 128 steps before decreasing to 64.0% at 2048 steps. Lastly, even Average merging which seemed more robust to the negative effects of overtraining on model merging reaches a peak accuracy of 59.3% at 128 steps before decreasing to 58.6% at 2048 steps.

# F    EFFECT OF MODEL SCALE

In the right panel of Figure 2, we have investigated how model size influences the merging dynamics by comparing Task Arithmetic merging results across T5-Base (220M parameters), T5-Large (770M), and T5-3B (3B). In Figure 8 we also show the average expert accuracy on their respective tasks as well as the merging accuracies for Average, Task Arithmetic and TIES methods. The purpose of these experiments is to test whether the decrease in merging accuracy observed after extended fine-tuning in smaller models also occurs at larger scales. We find that the same phenomenon persists: for both Task Arithmetic and TIES, merging accuracy peaks at an intermediate number of training steps and then degrades as fine-tuning continues, even though the absolute merging accuracy is generally higher for the larger models. Interestingly, Average merging appears robust to this degradation, but its overall accuracy remains comparatively low.

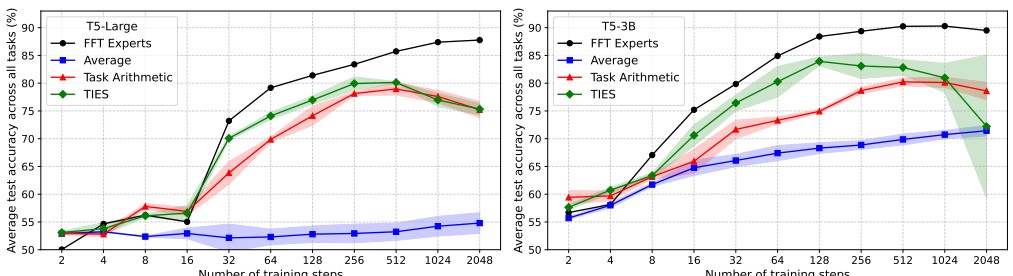

Figure 8: Average test accuracy across all 7 NLP tasks for fully fine-tuned T5-Large (**left**) and T5-3B (**right**) models. We plot the average accuracy of the expert models evaluated on their respective tasks as well as merging accuracies for multiple methods. Shaded regions show mean±std over 3 random seeds.

# G    EFFECT OF LORA RANK

In this section, we examine how the choice of LoRA rank affects the degradation effect reported in the main paper. We find that increasing the LoRA rank mitigates the loss in merging accuracy that occurs as experts are trained for longer.

We fine-tune ViT-B-32 models on the eight image-classification tasks from Section 2.5, applying LoRA adapters to every linear layer while systematically varying the adapter rank $r$. We employ square-root scaling for the LoRA factor $\alpha$, choosing $(r, \alpha) \in \{(16, 45), (32, 64), (64, 90), (128, 128), (256, 181)\}$. The models are trained for varying number of steps $s \in \{8, 32, 128, 512, 2048\}$ to assess how training duration interacts with rank. When merging, we combine LoRA-adapted models with the same rank and trained for the same number of steps. The resulting accuracies are plotted in Figure 9.

Across all three merging methods (Average, Task Arithmetic, and TIES) increasing the LoRA adapter rank consistently raises merging accuracy at every training duration. Moreover, higher ranks substantially attenuate the accuracy drop associated with extended training: as the number of fine-tuning steps grows, models with larger ranks exhibit smaller declines from their peak merging performance.

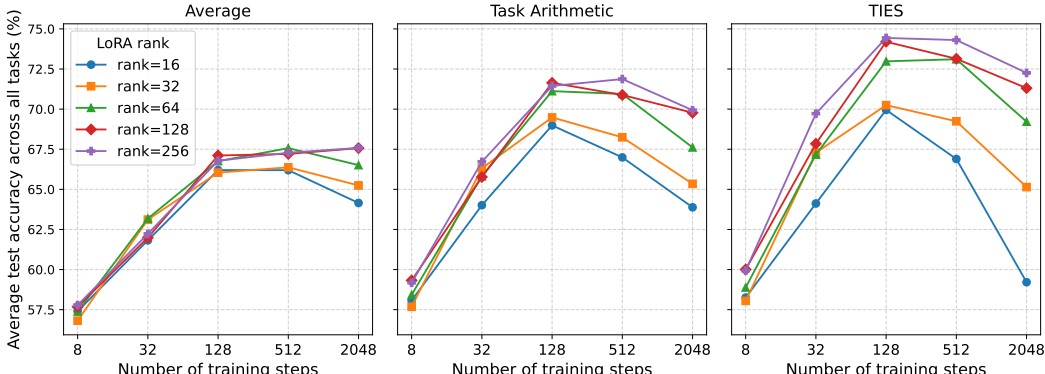

Figure 9: Average test accuracy across all 8 vision classification tasks as a function of the number of fine-tuning steps for different LoRA ranks and three merging methods. Each panel shows one method: Average (left), Task Arithmetic (center) and TIES (right). Colored solid lines and distinct markers denote the different LoRA adapter ranks. The x-axis is in $\log_2$ scale.

## H    MEMORIZATION ANALYSIS OF EARLY VS. LATE CHECKPOINTS

A central hypothesis in our work is that *longer finetuning drives models to increasingly memorize difficult training examples*. To test this, we compare CLIP ViT-B-32 models trained for **256** steps ("early") to models trained for **2048** steps ("late") across eight image classification datasets (Cars, DTD, EuroSAT, GTSRB, MNIST, RESISC45, SUN397, SVHN) with three random seeds each. Although both checkpoints show high training accuracy overall (100% on easy examples at both stages; 75.1% $\rightarrow$ 95.6% on hard examples from 256 to 2048 steps), memorization manifests not primarily in raw accuracy but in the *trajectory of prediction confidence, loss, and probability distributions*. We show that while accuracy on hard examples improves by 20.5 percentage points, the underlying memorization metrics exhibit changes that are *orders of magnitude* larger relative to easy examples, revealing the mechanism by which extended training drives memorization.

To quantify this, we compute three per-example metrics designed to detect late-stage memorization effects. All metrics are computed on the training set, and then aggregated into 10 difficulty bins per dataset using EL2N scores (Section 4). Each dataset is partitioned into 10 quantile-based bins, where bin 1 contains the easiest 10% of examples and bin 10 contains the hardest 10%. To prevent large datasets from dominating the averages, all results are first aggregated *per dataset* and then averaged across datasets (mean-of-means). For a training example with label $y$ at training step $t \in \{256, 2048\}$, we denote the predicted probability vector as $p^{(t)} = \text{softmax}(z^{(t)})$ where $z^{(t)}$ are the logits.

### H.1    MEMORIZATION METRICS

**Change in margin:**    $\Delta m = m^{(2048)} - m^{(256)}$ where $m^{(t)} = p_y^{(t)} - \max_{c \neq y} p_c^{(t)}$.
The margin measures the confidence gap between the true class probability and the maximum probability assigned to any other class. A strong positive relationship between $\Delta m$ and difficulty indicates that the late model becomes disproportionately confident on hard examples.

**Change in loss:**    $\Delta \ell = \ell^{(256)} - \ell^{(2048)}$ where $\ell^{(t)} = -\log p_y^{(t)}$.
This measures the drop in cross-entropy loss from early to late training. Large positive values indicate that the late model "forces down" the loss of examples that earlier checkpoints still struggled with, even when the top-1 prediction was already correct.

**Predictive distribution shift:**    $\text{shift} = \|p^{(2048)} - p^{(256)}\|_1$.
This $L_1$ distance quantifies how much the model's entire probability vector changes between checkpoints. Large shifts indicate substantial changes to the decision function, revealing targeted late-stage adjustments even when accuracy is unchanged.

## H.2 RESULTS

Table 4 summarizes the mean values of each metric for the easiest (bin 1) and hardest (bin 10) examples, averaged across all datasets and seeds. All three metrics display a strong positive correlation with difficulty bin number, indicating that additional training preferentially modifies the predictions of the hardest examples.

| Metric | Easy (bin 1) | Hard (bin 10) | Ratio (Hard/Easy) |
|---|---|---|---|
| $\Delta m$ (margin) | 0.0016 | 0.3911 | 244.6 |
| $\Delta \ell$ (loss) | 0.0009 | 0.5924 | 653.5 |
| Predictive shift ($L_1$) | 0.0039 | 0.5003 | 129.4 |

Table 4: Per-bin memorization indicators averaged across datasets and seeds. Hard examples exhibit dramatically larger changes between the 256- and 2048-step models, consistent with late-stage memorization.

## H.3 ANALYSIS

All three metrics exhibit a strong, monotonic increase across the 10 difficulty bins (Pearson correlations between bin number and per-bin metric values: $\Delta m$: $r=0.914$ ($p=0.0002$), $\Delta \ell$: $r=0.820$ ($p=0.004$), predictive shift: $r=0.920$ ($p=0.0002$)). These highly significant correlations demonstrate that difficulty bin systematically predicts the magnitude of model changes during extended training, ruling out random variation as an explanation. This pattern reveals a coherent story:

- **Confidence increases (margin)** for hard examples are more than two orders of magnitude larger than for easy examples ($244.6\times$). Easy examples already achieve near-maximal margins at 256 steps ($m \approx 0.997$), leaving little room for improvement, while hard examples undergo substantial margin increases ($\Delta m = 0.39$) as the model learns to confidently classify them.

- **Loss reductions** from 256 to 2048 steps show the most extreme differential effect ($653.5\times$), indicating that extended training "forces" difficult examples into the correct class by dramatically reducing their cross-entropy loss, even when the top-1 prediction was already correct at the earlier checkpoint.

- **Probability distributions** shift substantially more for difficult examples ($129.4\times$), suggesting targeted late-stage adjustments to the decision function. While easy examples maintain stable predictions ($L_1$ shift $\approx 0.004$), hard examples experience large redistributions of probability mass across classes ($L_1$ shift $\approx 0.50$).

Together, these metrics provide consistent and quantitative evidence that longer finetuning causes the model to memorize difficult examples: the 2048-step models exhibit large, difficulty-dependent changes to losses, margins, and probability distributions that are absent or minimal in the 256-step models. These memorization patterns have important implications for task vector composition and model merging. Models at different training stages have encoded fundamentally different solutions to the same classification task: early models rely on generalizable features that work across many examples, while late models additionally employ example-specific adjustments that memorize individual difficult cases. When such heterogeneous models are combined via task arithmetic or model merging, the interaction between generalizable and memorized components can produce unexpected emergent behaviors, potentially explaining performance variability in multi-task and few-shot transfer settings.

# I QUANTIFYING PARAMETER INTERFERENCE THROUGHOUT TRAINING

**Task Vector Construction** For each dataset $d$ and training step $t$, we compute a task vector $\tau_d^t$ as the parameter difference between the fine-tuned model and the pre-trained (zero-shot) model:

$$\tau_d^t = \theta_d^t - \theta_0 \tag{3}$$

where $\theta_d^t$ represents the parameters of the model fine-tuned on dataset $d$ for $t$ steps, and $\theta_0$ represents the pre-trained model parameters. All floating-point parameters are concatenated into a single vector of dimension $n$.

**Top-$k\%$ Pruning** For analysis, we apply magnitude-based pruning to the task vectors to retain only the most significant parameter changes. For a given task vector $\tau$ and pruning threshold $k\%$, we define the pruned task vector:

$$\tilde{\tau}_i = \begin{cases} \tau_i & \text{if } |\tau_i| \geq \text{threshold}_k \\ 0 & \text{otherwise} \end{cases} \tag{4}$$

where $\text{threshold}_k$ is the $k$-th percentile of $|\tau|$. Unless otherwise specified, we use $k = 20\%$ (retaining the top 20% of parameters by absolute magnitude), following the best practice from Yadav et al. (2023).

## I.1 METRIC DEFINITIONS

**Sign Conflict Percentage.** For each dataset pair $(d_1, d_2)$, we count the proportion of parameters where the task vectors have opposite signs. Let $\mathcal{A} = \{i : \tilde{\tau}_{d_1,i} \neq 0 \vee \tilde{\tau}_{d_2,i} \neq 0\}$ be the set of active parameters, and $\mathcal{C} = \{i \in \mathcal{A} : (\tilde{\tau}_{d_1,i} > 0 \wedge \tilde{\tau}_{d_2,i} < 0) \vee (\tilde{\tau}_{d_1,i} < 0 \wedge \tilde{\tau}_{d_2,i} > 0)\}$ be the set of parameters with conflicting signs. Then:

$$\text{SignConflict}(d_1, d_2) = \frac{|\mathcal{C}|}{|\mathcal{A}|} \times 100 \tag{5}$$

High sign conflict indicates destructive interference when averaging task vectors.

**Parameter Overlap Percentage** We measure how much the important parameters (top-$k\%$ by magnitude) overlap between datasets. For each dataset $d$, let $\mathcal{M}_d = \{i : |\tau_{d,i}| \geq \text{threshold}_k\}$ be the set of important parameter indices. The overlap between two datasets is:

$$\text{Overlap}(d_1, d_2) = \frac{|\mathcal{M}_{d_1} \cap \mathcal{M}_{d_2}|}{|\mathcal{M}_{d_1}|} \times 100 \tag{6}$$

High overlap indicates that different tasks compete for the same parameters, increasing the potential for interference during merging.

**Magnitude Ratio** For parameters that are important in both task vectors (i.e., in the overlapping set $\mathcal{O} = \mathcal{M}_{d_1} \cap \mathcal{M}_{d_2}$), we measure the disagreement in their magnitudes:

$$\text{MagRatio}(d_1, d_2) = \frac{1}{|\mathcal{O}|} \sum_{i \in \mathcal{O}} \frac{\max(|\tau_{d_1,i}|, |\tau_{d_2,i}|)}{\min(|\tau_{d_1,i}|, |\tau_{d_2,i}|)} \tag{7}$$

A magnitude ratio significantly greater than 1 indicates that even when tasks modify the same parameters, they disagree substantially on the extent of modification, leading to interference when averaged.

**Per-Parameter Variance** Unlike the pairwise metrics above, per-parameter variance captures multi-task disagreement. For each parameter position $i$, we compute the variance of its changes across all $D$ datasets:

$$\text{Var}(i) = \frac{1}{D} \sum_{d=1}^{D} (\tilde{\tau}_{d,i})^2 \tag{8}$$

where we assume the mean is centered at zero (the pre-trained model). The overall variance metric is:

$$\text{Variance} = \frac{1}{|\mathcal{A}_{\text{all}}|} \sum_{i \in \mathcal{A}_{\text{all}}} \text{Var}(i) \tag{9}$$

where $\mathcal{A}_{\text{all}} = \bigcup_{d=1}^{D}\{i : \tilde{\tau}_{d,i} \neq 0\}$ is the union of all active parameters across datasets. Higher variance indicates greater disagreement about how each parameter should be modified, leading to information loss during averaging.

## I.2   RESULTS

We analyze ViT-B-32 checkpoints trained on the 8 considered vision datasets (Cars, DTD, EuroSAT, GTSRB, MNIST, RESISC45, SUN397, SVHN) across 6 training step counts (4, 16, 64, 256, 512, 2048) and 3 random seeds.

For pairwise metrics (Sign Conflict, Parameter Overlap, Magnitude Ratio), we:

1. Compute the metric for all $\binom{8}{2} = 28$ dataset pairs at each step and seed
2. Average across the 28 pairs to obtain a single value per step per seed
3. Compute the mean and standard deviation across the 3 seeds

For the per-parameter variance metric, we:

1. Compute variance across all 8 datasets simultaneously at each step and seed
2. Compute the mean and standard deviation across the 3 seeds

The results are presented in Table 5 and Figure 10.

Table 5: Parameter interference metrics across training steps (mean $\pm$ std across 3 seeds). Metrics computed on the union of each dataset's top 20% parameters by magnitude.

| Steps | Sign Conflict (%) | Param Overlap (%) | Mag Ratio | Variance |
|---|---|---|---|---|
| 4 | $7.24 \pm 0.03$ | $26.09 \pm 0.07$ | $1.03 \pm 0.00$ | $(1.35 \pm 0.00) \times 10^{-10}$ |
| 16 | $7.20 \pm 0.01$ | $26.46 \pm 0.02$ | $1.23 \pm 0.00$ | $(1.06 \pm 0.00) \times 10^{-9}$ |
| 64 | $7.42 \pm 0.01$ | $27.67 \pm 0.00$ | $1.38 \pm 0.00$ | $(4.05 \pm 0.01) \times 10^{-9}$ |
| 256 | $7.77 \pm 0.01$ | $28.75 \pm 0.02$ | $1.43 \pm 0.00$ | $(1.27 \pm 0.01) \times 10^{-8}$ |
| 512 | $7.92 \pm 0.01$ | $29.08 \pm 0.02$ | $1.43 \pm 0.00$ | $(2.37 \pm 0.01) \times 10^{-8}$ |
| 2048 | $8.03 \pm 0.00$ | $29.53 \pm 0.01$ | $1.48 \pm 0.01$ | $(8.15 \pm 0.04) \times 10^{-8}$ |

Table 5 presents parameter interference metrics across training steps, revealing systematic increases in all interference measures as training progresses. Sign conflict percentage increases modestly from 7.24% to 8.03%, indicating a slight rise in parameters with opposing signs across tasks. Parameter overlap grows from 26.09% to 29.53%, showing that longer training causes different tasks to increasingly compete for the same important parameters. The magnitude ratio increases from 1.03 to 1.48, demonstrating growing disagreement about the extent of parameter modifications even when tasks modify the same parameters in the same direction. Most dramatically, per-parameter variance increases by approximately 60-fold from $1.35 \times 10^{-10}$ to $8.15 \times 10^{-8}$, providing strong evidence that extended training causes datasets to diverge substantially in their parameter modifications. Collectively, these metrics demonstrate that longer training amplifies parameter interference, directly explaining the degradation in merge performance observed with increased training steps. The consistency of these trends across all metrics and the low standard deviations across seeds indicate that this phenomenon is robust and reproducible.

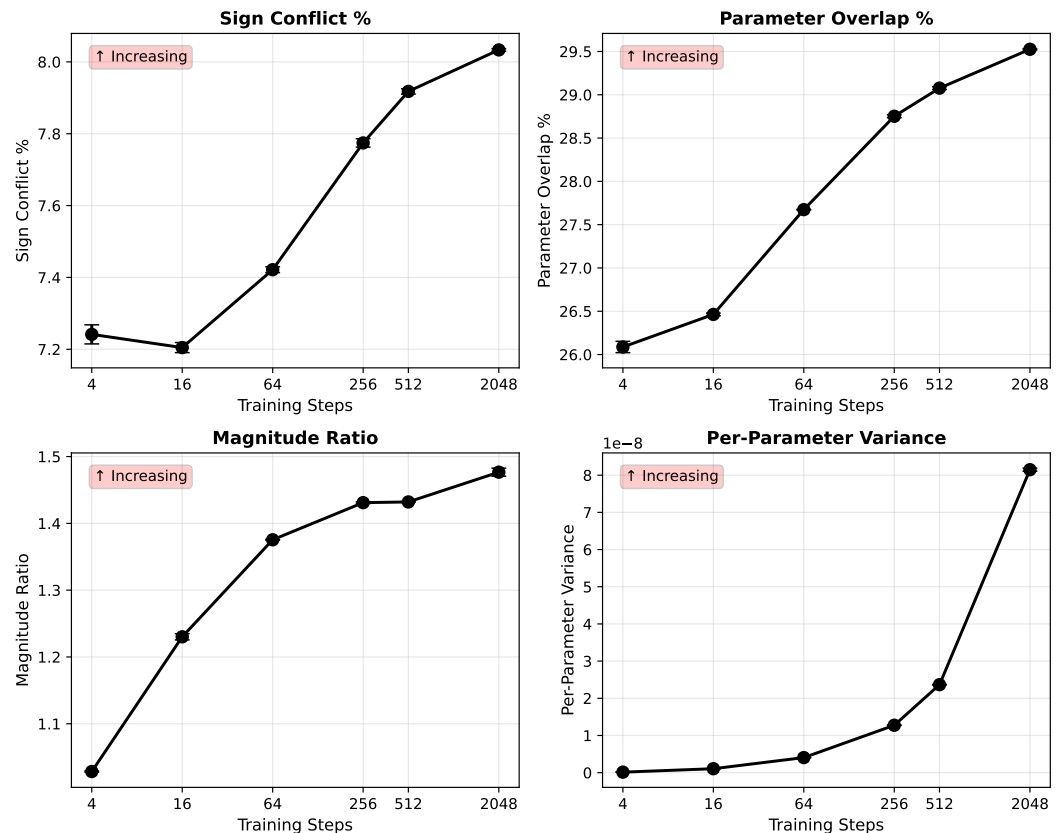

Figure 10: Parameter interference metrics across training steps (mean $\pm$ std across 3 seeds). Metrics computed on the union of each dataset's top 20% parameters by magnitude.

## J PSEUDOCODE FOR THE VIT EARLY STOPPING STRATEGY

Here we provide pseudocode for the LR scheduler and early stopping strategy with linear warm-up and adaptive decay used for ViT models.

**Algorithm 1:** Early stopping strategy with linear warm-up and adaptive decay

---

**Input:** Peak LR $\eta_{\max}$, warm-up steps $S_{\text{warm}}$=50, validation interval $S_{\text{val}}$=5, patience $P$=3, decay factor $\gamma$=0.5, minimum LR threshold $\eta_{\min} = 1e-7$

Initialize learning rate $\eta$=0, best validation accuracy $A_{\text{best}}$=0, plateau counter $c$=0

**for** *training step* $s = 1, 2, \ldots$ **do**

  **if** $s \leq S_{warm}$ **then**

                                                    // Linear warm-up

    $\eta \leftarrow \eta_{\max} \cdot \frac{s}{S_{\text{warm}}}$

  **else**

                            // Adaptive decay and early stopping

    **if** $s \bmod S_{val} = 0$ **then**

      Evaluate validation accuracy $A_{\text{val}}$

      **if** $A_{val} > A_{best}$ **then**

        $A_{\text{best}} \leftarrow A_{\text{val}}$

        $c \leftarrow 0$

      **else**

        $c \leftarrow c + 1$

      **if** $c \geq P$ **then**

        $\eta \leftarrow \gamma \cdot \eta$

        $c \leftarrow 0$

      **if** $\eta < \eta_{\min}$ **then**

        **stop training**

  Update model parameters using $\eta$

---

