# OpenReview forum: "Less is More: Undertraining Experts Improves Model Upcycling"
_ICLR.cc/2026/Conference — Submitted to ICLR 2026_

### Official Review · Reviewer_AM5C · 2025-10-31

**Soundness:** 3
**Presentation:** 3
**Contribution:** 3
**Rating:** 6
**Confidence:** 4

**Summary:**

This paper challenges the assumption that stronger task-specific "experts" that are tuned on top of a base model lead to stronger upcycled models (weight-merged or MoE-upcycled). To this end, authors explore the transfer between expert training and upcycling with respect to broader capabilities and knowledge transfer.

They find that long fine-tuning (referred to as "over-training"), either via PEFT or full fine-tuning, can hurt model upcycling for both model merging and MoE initialization. The paper offers an analysis using data difficulty and shows that easy data samples are correctly classified, but the upcycled models suffer from forgetting the hard data points. Finally, the paper shows that task-specific training time as a form of early-stopping can mitigate the negative impact of the observed phenomenon and recover optimal upcycling performance.

**Strengths:**

1. This paper is very well written. It presents the claim in a clean way and supports the argument in a large set of experiments, including 2 finetuning strategies, 2 upcycling methods (merging and MoErging), and using 2 domains (vision and language).

2. Analysis of observed results by using data difficulty shows good insights and explains some of the findings.

3. The paper proposes a simple but effective early-stopping method for better upcycling.

**Weaknesses:**

1. The only main weakness of the paper is that task averages might be misleading some of the results, whether some particular tasks are making the drops more significant. This is valid for both expert results and results for upcycled models. A closer look at the tasks and their results would be good.

2. The paper is very insightful as it shows an important phenomenon, but as mentioned in the paper, works like (Pari et al, 2024) also make quite similar observations, which slightly limits the novelty.

**Questions:**

1. For LoRA, optimal expert performance looks much closer to optimal upcycling performance according to Figures 1 and 2, suggesting that high over-optimization to the target task without marginal gain, which is slightly different than full finetuning. Based on that, more details on the difference between LoRA vs full finetuning would be nice.

2. The analysis showing that difficult examples are being forgotten after upcycling is very informative. Since line 378 (page 8) suggests a sweet spot for memorization, did you check the degree of memorization of such hard examples in early checkpoints?

---

> ### Author Response · Authors · 2025-11-26
>
> We thank the reviewer for their thoughtful and positive assessment of our work, and for highlighting the clarity of our exposition and the breadth of our experimental validation. We are glad the data difficulty analysis and early-stopping strategy were found insightful. We address each comment and question below.
> ### Weakness 1: Concern about task averages potentially hiding per-task variation
> This is a valid concern. However, the per-task breakdowns (now added in Appendix D for both vision and NLP settings) confirm the phenomenon is systematic, not outlier-driven. While there is some variability in the magnitude of the overtraining effect across datasets and merging methods, the direction is consistent: for most tasks and methods, experts trained for significantly fewer steps yield higher merging accuracy. We do not rely on any single dataset or outlier task to support our conclusions.
> ### Weakness 2: Novelty relative to past works
> Prior works observe related phenomena but do not validate or analyze them in depth. (Pari et al., 2024) identify representational incompatibility (via CKA) in a small, two-expert setting and propose bypassing merging altogether in favor of routing. (Zhou et al., 2025) observe that early task vectors improve task arithmetic, but do not analyze why; they propose alternating training and merging steps, which is unrealistic in most merging scenarios and closer to distributed training.
> Our contributions go substantially beyond these observations:
> 1. **Validation across settings:** We demonstrate the phenomenon across 2 fine-tuning strategies, 2 upcycling methods (merging and MoErging), and 2 domains/architectures (vision and language).
> 2. **Mechanistic explanation:** We explain the phenomenon via the memorization of hard examples, linking our results to the rich literature on data difficulty..
> 3. **Practical mitigation:** We provide a simple, effective solution through early stopping.
>
> The breadth of validation, the mechanistic grounding, and the actionable mitigation distinguish our work from prior observations.
>
>
> ### Question 1: LoRA vs FFT
> The reviewer states that "For LoRA, optimal expert performance looks much closer to optimal upcycling performance." However, our results show the opposite: the gap between maximum expert accuracy and merged model accuracy is substantially *larger* for LoRA than for FFT. In the vision setting (Fig. 1), experts reach ~90% accuracy in both cases, but FFT merges reach ~75% while LoRA merges reach less than 70%. The same pattern holds in the NLP setting (Fig. 2). If we have misunderstood the reviewer's point, we welcome clarification.
>
> ### Question 2: Memorization
> Quantifying the absolute degree of memorization in a given model is difficult, however comparative analysis is more straightforward. We have added results in Appendix H studying memorization in early (256 steps) vs. late (2048 steps) checkpoints. Both checkpoints achieve close to 100% accuracy on easy examples. On the most difficult test examples, accuracy increases from 75.1% at 256 steps to 95.6% at 2048 steps. More tellingly, relative quantities, change in margin (confidence gap between true class probability and maximum probability across other classes), change in loss, and predictive distribution shift (L1 distance between probability vectors) are orders of magnitude higher for the hardest examples than for the easiest examples. This confirms that hard-example memorization is low in early checkpoints and increases dramatically during later training stages, consistent with our analysis. These results are discussed in the revised Section 4.
>
> ### Conclusion
> We thank the reviewer for their thoughtful and constructive feedback. The revisions. including additional experiments in Appendices D and H and clarifications in Section 4, address all raised concerns. We appreciate the reviewer's engagement and welcome any further discussion.

---

### Official Review · Reviewer_qBGP · 2025-10-31

**Soundness:** 3
**Presentation:** 4
**Contribution:** 3
**Rating:** 6
**Confidence:** 4

**Summary:**

The paper investigates how expert fine-tuning affects model upcycling—specifically, model merging and MoErging (mixture-of-experts merging). The authors ask two key questions: (1) How does expert training influence upcycling performance? and (2) Do all capabilities and knowledge transfer equally well? Through comprehensive experiments on both vision (CLIP-ViT-B-32) and language (T5) domains, the paper demonstrates that longer fine-tuning (overtraining) of experts can harm merging and MoErging performance. The authors attribute this to the memorization of a small set of difficult examples that are subsequently forgotten during merging. They propose an aggressive early-stopping strategy that improves upcycling outcomes and validate it across multiple settings, including LoRA-based and fully fine-tuned experts. The inclusion of MoE evaluations further strengthens the paper’s empirical scope.

**Strengths:**

- The paper presents a well-designed empirical study with clear hypotheses and extensive experiments across vision and language domains.

- Novel perspective: It challenges a strong but untested assumption in model upcycling research and provides evidence for the benefits of undertraining.

- Clarity and reproducibility: The methodology, datasets, and hyperparameters are clearly described, and the open-source implementation is a strong plus.

- The inclusion of MoE evaluations adds depth and relevance, demonstrating the findings’ applicability to modular architectures.

- The early stopping strategy is simple and effective

**Weaknesses:**

* The main limitation, also acknowledged by the authors, is the lack of actionable implementation details. The paper recommends publishing intermediate checkpoints and applying early stopping, but does not specify how many checkpoints to release, how to store or curate them, or how early stopping parameters should vary across tasks or domains.
* Experimental diversity: The study relies on single model families in each domain (ViT for vision and T5 for NLP). Evaluating additional architectures (e.g., ViT-L, BERT, or OPT) would better establish generalization.
* The analysis in Section 5 (L415–L429) is dense; integrating the early-stopping algorithm in pseudocode could enhance readability.
* Section 2.4 could be reordered for clarity, introducing the chosen difficulty score (EL2N) before citing the broader literature would improve readability.
* Some redundancy exists between Related Work and Preliminaries; consolidating them could tighten the paper.
* While the empirical results are compelling, a more formal or theoretical treatment (e.g., why memorization of hard examples leads to poor parameter compatibility) would strengthen the contribution.
* Minor issue: Figure 4 omits DARE results without explanation. Please clarify.

**Questions:**

* Why is DARE missing from Figure 4?

* Could you extend experiments to include two or more models per domain to demonstrate generalization?

* Can the authors quantify how many intermediate checkpoints are typically optimal for upcycling workflows?

* Do the authors expect these findings to generalize to multi-modal or regression tasks?

*  I would suggest working on how you can extract actionable insights from the current work; this can strengthen the work.

---

> ### Author Response · Authors · 2025-11-26
>
> We thank the reviewer for the thoughtful and constructive feedback. We are encouraged by the positive assessment of the soundness, contribution, and clarity of our work. We address all concerns below.
>
> ### **Lack of actionable insights** (weakness 1 & questions 3 and 5)
> Our early stopping results provide direct, actionable insights regarding the publishing of intermediate checkpoints for upcycling workflows. They show that stopping training when the accuracy on a held-out validation set starts to plateau yields better upcycling results than completing training, even if expert accuracies are lower. Therefore, we propose a concrete change to the standard model release protocol based on our findings. Practitioners should not only release the final converged model but an additional intermediate checkpoint extracted during training as soon as the validation accuracy starts plateauing. The sharing of this single intermediate checkpoint is a low-cost modification to training pipelines that can significantly help downstream upcycling applications. This intermediate checkpoint can be shared just like the final one, on platforms such as HuggingFace and doesn't require any special curation. We’ve made this suggestion explicit in the conclusion.
>
> ### **Experimental diversity** (weakness 2 & question 2)
> To definitively address the generalization concern, we have run entirely new experiments with ViT-Large (Vision) and BERT (NLP) models during the rebuttal period, as per the reviewer’s request. The results (now in Appendix E) exhibit similar merging degradation curve as the ViT-B and T5 models. We’ve now established the existence of this phenomenon across domains (vision and NLP), model families and sizes, fine-tuning strategies (FFT and LoRA) and across upcycling strategies (merging and MoErging). This confirms that the interference driven by hard-sample memorization is a fundamental property, not an artifact of specific architectures.
>
> ### **Dense analysis in Section 5** (weakness 3)
> We have added pseudocode summarizing the early-stopping strategy for the vision setting in Appendix J. This should make Section 5 more accessible.
>
> ### **Clarity of Section 2.4** (weakness 4)
> We have updated Section 2.4 so that EL2N is introduced before discussing related data difficulty literature and have lightly reworked the text for clarity.
>
> ### **Redundancies between Related Work and Preliminaries** (weakness 5)
> We have revised both sections to clearly separate foundational background from directly comparable prior work. In particular, we have moved the general discussion of model merging and model MoErging, including their methodological context and broad literature, to the Preliminaries (Sec. 2), since these concepts are integral to our setup. The Related Work section now focuses exclusively on prior studies that are directly comparable to our contributions.
>
> ### **Theoretical explanation of results** (weakness 5)
> A full theoretical analysis lies outside the scope of this paper. However, we provide an intuitive mechanistic explanation through our data difficulty and training dynamics analysis. As shown in Fig. 3 (left), the gradient signal in late-stage fine-tuning becomes dominated by a small fraction of the hardest training examples, which prior work on data difficulty (Sec. 2.4) shows often contain task-specific, outlier, or noisy features. Learning these features improves single-task accuracy but induces parameter updates that diverge across experts, reducing their compatibility for merging. We have added results in Appendix I linking longer training to increased parameter interference. Undertrained experts, by contrast, have not yet entered this high-divergence regime or have spent less time in it: their updates are shaped by a broader subset of examples with more shared structure across tasks, yielding better cross-task agreement and more mergeable parameters. Our results thus link the well-known issue of parameter interference in model merging to the concrete training dynamics of hard-example memorization, explaining why overtraining yields marginal expert gains that not only fail to transfer through merging but actively harm it. We have incorporated this explanation into Section 4.
>
> ### **Missing DARE results in Fig. 4** (weakness 6 and question 1)
> We have added the DARE results to Fig. 4. The same phenomenon is observed: the 30% most difficult examples represent over 50% of forgotten examples. This additional method strengthens our conclusion and its generality.

---

> ### Author Response · Authors · 2025-11-26
>
> ### **Generalization to multi-modal or regression tasks** (question 4)
> We expect our findings to generalize. The core mechanism we identify, late-stage training being dominated by difficult examples that induce idiosyncratic, task-specific parameter updates, is not specific to classification or single-modality settings. Data difficulty is a general property: multi-modal datasets will contain examples that are harder to align across modalities, and regression tasks will have high-residual outliers that dominate late-stage gradients. The fundamental tension between single-task optimization (which benefits from memorizing difficult examples) and cross-task compatibility (which suffers from the resulting parameter divergence) should persist across task types.
>
> We thank the reviewer for their constructive engagement and thoughtful questions. We have revised the manuscript to clarify the mechanisms underlying our findings and to include additional experiments that reinforce our claims. We believe these revisions address all raised concerns, and we welcome any further discussion.

---

### Official Review · Reviewer_HU9q · 2025-11-01

**Soundness:** 3
**Presentation:** 3
**Contribution:** 2
**Rating:** 2
**Confidence:** 4

**Summary:**

The paper notes that the long fine-tuning of experts experts leads to the degradation in the performance of merged models, as the late part of the fine-tuning process focuses on the difficult samples. Thus, the paper proses a task-dependent early stopping strategy to improve the performance of a merged model.

**Strengths:**

- The paper introduces a different perspective/new insight on why the further training of expert hurts the overall performance.
- The paper shows that LoRA trained for as few as 4 steps leads to better performance than overtrained experts after merging. Further, the paper shows that training an expert for only 1/8 of training iterations leads to better performance.
- The paper is well-written and easy to follow.

**Weaknesses:**

- The result that demonstrates the benefit of undertraining in the context of model merging.
- The paper lacks discussions on other works that analyze the influence of fine-tuning stage on the performance of merged models [A,B,C,D].
- The paper proposes to reduce LR on plateau, which however is not new, but rather, a common technique.
- It is a bit hard to see why reducing LR on plateau is an effective early stopping strategy for model upcycling.
- The strategy of reducing LR on plateau has weak connections with the paper's perspective and analysis on hard examples being dominant loss signal during the later stage of the fine-tuning.
- The paper claims,  "memorizing difficult samples, with uncommon features or noisy labels likely to yield parameter updates which are unique from one dataset to the other, and which will be destroyed by the aggregation step of model merging." But, this seems to contradict the another claim by the paper: difficult samples are still beneficial to the generalization, when the paper expresses such difficult samples as samples with uncommon features or noisy labels. Furthermore, is there any theoretical/empirical/intuitive explanation why the knowledge of difficult samples is destroyed during the merging?
- The perspective on how overtraining/focusing on difficult samples has adverse influences on model upcycling does not seem to be too different from the common explanation that the performance degradation of a merged model comes from the parameter interference. Easy samples can be considered to contain similar knowledge to the pretrained model's prior knowledge. Meanwhile, difficult samples contain more task-specific and hence different knowledge from the pretrained model's prior knowledge. Thus, further training on each task naturally leads to more parameter interference or conflict between experts.

[A] Jin et al., Fine-Tuning Attention Modules Only: Enhancing Weight Disentanglement in Task Arithmetic. ICLR, 2025.
[B] Lee et al., Mitigating Parameter Interference in Model Merging via Sharpness-Aware Fine-tuning. ICLR, 2025.
[C] Tang et al., Parameter Efficient Multi-task Model Fusion with Partial Linearization. ICLR, 2024.
[D] Ortiz-Jimenez et al.,Task Arithmetic in the Tangent Space: Improved Editing of Pre-Trained Models. NeurIPS, 2023.

**Questions:**

- Why are difficult samples forgotten during merging, even though the late part of fine-tuning process is mostly driven by difficult samples?
- Why is reducing LR on plateau effective strategy for model merging? How is this related to the phenomenon in which fine-tuning process focuses on difficult samples during the later stage?

---

> ### Author Response · Authors · 2025-11-26
>
> We thank the reviewer for recognizing the novelty of our perspective, the clarity of the paper, and the strength of our empirical findings. We address each concern below.
> ### On the results demonstrating the benefit of undertraining in the context of model merging
> The reviewer listed this as a weakness but did not elaborate, despite also highlighting several of these results as strengths. We note that these findings are both robust and meaningful: across domains (vision and NLP), architectures (ViT, T5 and newly added BERT), and fine-tuning regimes (FFT and LoRA), we observe substantial and consistent drops in merging accuracy when experts are overtrained. Notably, overtraining can have a larger negative effect on merging accuracy than the choice of merging method itself (line ~290). While recent works focus on improving merging algorithms, very few have examined how fine-tuning duration affects mergeability. Our results fill this gap and demonstrate that training dynamics play a central role in upcycling performance.
> ### Missing discussion of related works
> We are aware of works [A, B, C, D]. However, these are orthogonal to our contribution: [A] fine-tunes only specific submodules, [B] employs sharpness-aware optimization, and [C, D] rely on linearization-based approaches. Our goal is to analyze standard fine-tuning protocols without merging-specific adjustments, as these form the baseline for most upcycling pipelines and represent the vast majority of open-source models (e.g., on HuggingFace). Our work reveals why this standard ecosystem can yield sub-optimal upcycling results, a question the cited works do not address. We have added these works to the Related Work section to clarify this distinction.
> ### Reducing the LR on plateau & New early stopping results
> The reviewer raised concerns regarding our “reduce LR on plateau” strategy, specifically regarding its novelty, its effectiveness, and its connection to our hard-example analysis (weaknesses 3, 4, 5 and question 2). Since these comments are closely related, we address them together.
>
> **The novelty is not the scheduler; rather the novelty is the discovery that standard full convergence is harmful for upcycling.** Reducing LR on plateau is a standard technique; we do not claim it as a contribution. Our primary contribution is the systematic discovery that overtraining experts hurts upcycling performance. This core finding naturally motivates the use of **early stopping as a general principle to mitigate this effect**. Our goal was not to prove the absolute optimality of one specific strategy, but to propose a practical, automated, and effective one that builds on established best practices.
>
> Regarding why we chose the “reduce LR on plateau” variant specifically: our goal was to use an early-stopping mechanism that (i) behaves similarly to the cosine decay used in our vision experiments when early stopping is not applied, and (ii) adapts flexibly to different tasks, whose optimal stopping points vary substantially. Linear warm-up followed by reduce LR-on-plateau provides a straightforward way to introduce a task-dependent decay schedule that shortens training when progress slows.
>
> The reviewer asks why LR-on-plateau is effective. The answer follows directly from our analysis: standard training pursues minimal loss, which entails memorizing difficult examples with task-specific features. Upcycling requires feature sharing across tasks. Early stopping prevents the model from staying too long in the memorization regime we identified, it truncates the late-stage updates dominated by high-EL2N examples, precisely the updates most poorly preserved by merging. We have also added results in Appendix I showing that longer training systematically leads to increases in parameter interference, early stopping mitigates this. In this sense, LR-on-plateau is simply one practical instantiation of the broader principle that emerges from our analysis: any properly designed early-stopping strategy would have similar effects.
> We have updated the paper to distinguish between the general role of early stopping (central to our findings) and the particular implementation (not a claimed contribution).
>
> #### Additional results in the NLP setting
> To isolate the effect of early stopping from scheduler confounds in the vision setting, we conducted additional NLP experiments which are now in the paper. Our T5 models use a constant LR, enabling straightforward early stopping: we evaluate on a held-out validation set every `num_steps_eval` steps, save the model when validation accuracy reaches a new high, and stop if accuracy fails to improve for `num_evals_patience` evaluations. We tested two configurations and report results in Section 5. **The conclusions hold: early stopping yields significantly better merged models (particularly for TA and TIES) even when expert accuracies are comparable or lower.** In fact, early-stopping provides even larger gains than in the vision setting.

---

> > ### Author Response · Authors · 2025-11-26
> >
> > ## On the memorization of difficult samples and parameter interference
> > The reviewer raises two related concerns: (1) an apparent contradiction between memorization benefiting generalization yet harming merging, and (2) whether our perspective differs from standard parameter interference explanations. We address both, as they are closely linked.
> >
> > ### On the apparent contradiction
> > The statement that memorizing difficult samples is beneficial to generalization is not a contradiction. Data difficulty literature establishes that some memorization is necessary for close-to-optimal single-task generalization [1, 2, 3]. However, what benefits single-task performance can harm cross-task mergeability. Memorizing difficult examples improves individual expert accuracy, but the associated parameter updates are idiosyncratic and task-specific. When merged, these divergent updates cause parameter interference which is known to hurt model merging. We have added a brief discussion of this literature in Section 2.4 and have added results in Appendix I linking longer training to increased parameter interference.
> >
> > ### On the link to parameter interference
> > The reviewer suggests our perspective is not substantially different from standard parameter interference explanations. We point out that our perspective does not compete with the parameter interference one, rather they go hand in hand. Parameter interference is used to explain why merged models sometimes have lower accuracies than the experts being merged; our work provides a mechanistic explanation of why longer training leads to parameter interference through the lens of hard example memorization. Specifically:
> > 1. Late-stage training is dominated by difficult examples with task-specific or outlier features (Fig. 3)
> > 2. Learning these examples produces idiosyncratic parameter updates increasing parameter interference (new experiments in Appendix I)
> > 3. These updates are the primary drivers of interference during merging
> > 4. Early stopping mitigates this by truncating the high-divergence memorization regime from later training
> >
> > This understanding is actionable: it motivates early stopping as a principled strategy and predicts which samples will be forgotten (hard examples in Fig. 4). The standard parameter interference framing provides neither insight. Our contribution expands the interference explanation by grounding it in training dynamics and data difficulty.
> >
> > This also answers the reviewer's question "Why are difficult samples forgotten during merging?": it is precisely because late training is dominated by these examples that the resulting updates are idiosyncratic. Task-specific updates conflict across experts when merging, causing higher interference and forgetting for exactly these samples.
> > We have rewritten Section 4 to make this mechanism explicit.
> >
> > ### Conclusion
> > We have revised the manuscript to address these points. The revisions should fully address all raised concerns, and we welcome any further questions.
> >
> > ### References
> > [1] Idan Attias, Gintare Karolina Dziugaite, Mahdi Haghifam, Roi Livni, and Daniel M Roy. Informa-
> > tion complexity of stochastic convex optimization: Applications to generalization and memoriza-
> > tion. arXiv preprint arXiv:2402.09327, 2024.
> >
> > [2] Vitaly Feldman. Does learning require memorization? a short tale about a long tail. In Proceedings
> > of the 52nd Annual ACM SIGACT Symposium on Theory of Computing, pp. 954–959, 2020.
> >
> > [3] Vitaly Feldman and Chiyuan Zhang. What neural networks memorize and why: Discovering the
> > long tail via influence estimation. In H. Larochelle, M. Ranzato, R. Hadsell, M.F. Balcan, and
> > H. Lin (eds.), Advances in Neural Information Processing Systems, volume 33, pp. 2881–2891.
> > Curran Associates, Inc., 2020. URL https://proceedings.neurips.cc/paper_files/paper/2020/file/1e14bfe2714193e7af5abc64ecbd6b46-Paper.pdf

---

> ### Comment · Reviewer_HU9q · 2025-11-27
>
> The reviewer would like to thank the authors for the rebuttal.
> However, the concerns regarding the contribution of the paper remain.
> It is well known that deep learning models learn easy samples first and difficult samples later [E].
> The longer training of each expert leading to degradation in a merged model's performance does not come as a new finding, as it has been observed in previous works. For example, Figure 9 in [D] shows that the disentanglement error (one of measure for parameter interference) increases as parameters become closer to those of each task expert.
> And, the reviewer believes that proposed solution of early stopping is a standard regularization technique and choosing the number of training iterations is a matter of hyperparameter tuning.
>
> [E] Baldock et al., Deep Learning Through the Lens of Example Difficulty. NeurIPS, 2021.

---

> ### Author Response · Authors · 2025-12-01
>
> We thank the reviewer for their continued engagement. We respectfully address the remaining concerns.
>
> **On "easy-then-hard" learning being known**
>
> We agree [E] establishes that models learn easy examples before hard ones. However, this observation alone does not imply our findings. Our contribution is connecting this to upcycling: hard-example memorization induces idiosyncratic parameter updates that increase interference, causing systematic merging degradation—and the same hard examples are disproportionately forgotten (Figure 4). This causal chain linking data difficulty to parameter interference and upcycling outcomes has not been established before.
>
> **Longer training leading to degraded merging**
>
> As discussed in our related works section, prior works that touch on this phenomenon do not study it systematically. Zhou et al. (2025) only test 1-epoch fine-tuning; Pari et al. (2024) study only two-model merges and propose bypassing merging altogether. None establish generality across domains, architectures, fine-tuning regimes, and upcycling methods, nor provide mechanistic analysis linking training dynamics to parameter interference. Briefly noting a phenomenon is not equivalent to systematically studying it and providing an explanation.
>
> **On Figure 9 in [D]**
>
> Figure 9 in [D] shows disentanglement error as a function of the task vector scaling parameter ($\alpha$), not training duration. This measures how interference changes during the *merging process* as task vectors are scaled, which is fundamentally different from our analysis of how *training dynamics* affect the quality of experts being merged. [D] does not study training duration, does not connect interference to data difficulty, and provides no mitigation strategy beyond linearization. Our systematic study across 2 domains, 4 architectures, 2 fine-tuning regimes, and 2 upcycling methods—with mechanistic analysis (Appendices H, I)—goes well beyond this single figure.
>
> **On early stopping**
>
> We do not claim early stopping as a contribution. The contribution is identifying *why* it helps for upcycling (avoiding the high-interference memorization regime) and showing that checkpoint selection can matter more than merging algorithm choice. This is not hyperparameter tuning—it is identifying a previously unrecognized failure mode with explanation and mitigation.

---

### Official Review · Reviewer_LnVv · 2025-11-01

**Soundness:** 2
**Presentation:** 3
**Contribution:** 2
**Rating:** 2
**Confidence:** 3

**Summary:**

This paper investigates the relationship between expert model fine-tuning time and the subsequent model upcycling performance. The authors found that merging / moering with undertrained models lead to better performance, and propose to adopt a learning rate schedule with early stopping for model upcycling.

**Strengths:**

1. Writing is clear and easy to understand.
2. The experiments cover wide aspects, across both vision and NLP domain, model merging and model moering, as well as different architectures.

**Weaknesses:**

1. The explanation why overtraining hurts merging performance is unconvincing. The authors attribute that overtrained experts hurt merging performance because later training steps “memorize” difficult samples, which are later forgotten during the merging stage. However, this explanation at most explains why overtrained experts do not improve over undertrained experts, as the forgetting also happens with the undertrained experts as well (or it is even worse with undertrained experts, because according to the authors’ claim these hard samples were not even properly learned). As pointed out by the authors, the “overtraining” benefits single-task generalization, thus the real question here is why the improved generalization on single-task hurts performance of the merged model, which is not convincingly investigated.

2. The conclusion that “hard samples are learned later” seems to be using a circular logic. In the authors’ definition, the “hard samples” are the samples with low EL2N scores after 32 steps, i.e., they are the samples not properly learned in the early learning stage. Then naturally, by definition, these “hard samples” are learned later, which incurs a circular logic. The potential logic issue thus weakens the hypothesis and conclusion.

3. The early stopping strategy described in last section offers somewhat limited methodological innovation, as it is a standard learning rate schedule. There exist also the following issues:

a. Rather than just adding an early stopping, it changes also the learning rate schedule (from cosine decay to adaptive decay) compared to the baselines, mixing several confounders. Therefore, it is unclear the accuracy gains are brought by the learning rate schedule change or early stopping alone. Moreover, the lack of expert accuracy results for early-stopped models makes it difficult to assess whether early stopping truly prevents overtraining or simply reduces training.

b. Based on Figure 1 (left), suppose we apply early stopping on the original learning rate schedule, the training will not get stopped as the held-out-set accuracy has been improving throughout the training process. In that case, can early stopping still help?

**Questions:**

See weaknesses.

---

> ### Author Response · Authors · 2025-11-26
>
> We thank the reviewer for their kind words regarding the writing of our paper and the breath of our experiments. We address the identified weaknesses below.
>
> ### Weakness 1: “The explanation why overtraining hurts merging performance is unconvincing.”
> We respectfully disagree that our explanation as to why overtraining hurts model merging is insufficient. Our explanation is supported by numerous empirical results and aligns with both the model merging and data difficulty literatures. The reviewer asks "why the improved generalization on single-task hurts performance of the merged model". The answer is that **the memorization of hard examples which takes place during later stages of training yields idiosyncratic parameter updates that cause negative parameter interference and therefore hurt merging.**
>
> Our results are clear and we have added supporting experiments to make our conclusions even stronger:
>
> Later training stages are dominated by the memorization of hard examples, i.e. the learning of outlier / hard features. This is shown in Fig. 3 (left) with over 50% of the loss in mid and late training coming from the hardest 10% of data points. We have also added experiments in Appendix H showing that between 256 and 2048 steps, training yields only minor improvements in accuracy while the changes in margin, loss, and predictive distribution are orders of magnitude higher for hard examples than for easy examples, further supporting this claim. This aligns with past works establishing that some memorization is necessary for close-to-optimal generalization [3, 4, 5].
>
> Memorizing these hard examples implies learning uncommon features, yielding idiosyncratic parameter updates that do not generalize across tasks. We have added experiments in Appendix I showing that longer training systematically leads to increases in sign conflicts, parameter overlaps, magnitude ratios, and per-parameter variance between task vectors. This is consistent with the established understanding that parameter interference negatively affects merging performance.
>
> Crucially, undertrained experts haven't spent as much time in this "high-divergence" regime. Their parameter updates are the result of more balanced gradients coming from a broader set of examples with more shared structure across tasks. Thus:
> - Undertraining → fewer incompatible parameter updates → higher cross-task agreement → more mergeable
> - Overtraining → more task-idiosyncratic late-stage parameter updates → lower agreement between tasks → severely degraded by merging
>
> This directly explains why overtraining helps single-task accuracy but harms merging performance: the marginal accuracy gains at the end of training are driven by idiosyncratic features that not only cannot survive merging but also harm it. We have rewritten Section 4 to make this explanation clearer and have incorporated the newly added experiments to reinforce these claims.
>
> ### Weakness 2: “The conclusion that ‘hard samples are learned later’ is circular because difficulty is defined using EL2N at early steps.”
> Our claims **do not** rely on circular logic. The reviewer appears to equate "high EL2N score", i.e. hard example, with "learned later," which would make the conclusion tautological. This conflation is incorrect for the following reasons:
> - **Different models:** We calculate EL2N scores using 10 separate models (seeds 1–10) early in training (step 32), following best practices [1]. We then observe training dynamics on independent models (seeds 101–103) at step 2048. The measurement and observation are completely independent.
> - **Predictive, not definitional:** We do not define hard samples as those learned later. We define them via EL2N, which measures difficulty as an intrinsic characteristic of the data. We then discover empirically that these samples dominate the gradient signal in late training. This is a non-trivial empirical finding, not a definition.
> - **External validation:** As noted in Section 2.4, [2] shows that EL2N scores correlate strongly with other difficulty metrics which are *not computed early in training*, such as the diverse ensembles (DDD) score and the memorization score. This confirms that EL2N measures an intrinsic data property, not the order in which examples are learned.
> - **Independence:** Whether an example is "learned" is a model-data relationship, independent of its intrinsic difficulty. A high-EL2N example may be correctly classified by model A and misclassified by model B; this does not change its intrinsic difficulty.
>
> The fact that EL2N scores at step 32 predict which examples dominate loss 1000+ steps later, and that merging disproportionately forgets these same examples (Fig. 4), is evidence of a predictive relationship, not definitional circularity.

---

> > ### Author Response · Authors · 2025-11-26
> >
> > ### Weakness 3: Early-stopping
> >
> > We do not claim our early stopping strategy as a novel contribution. Our primary contribution is the systematic discovery that overtraining experts hurts upcycling performance. This core finding naturally motivates the use of **early stopping as a general principle to mitigate this effect**. Our goal was not to prove the absolute optimality of one specific strategy, but to propose a practical, automated, and effective one that builds on established best practices.
> >
> > We address the specific concerns below:
> > * **a) Learning rate schedule changes:** The reviewer notes that our early stopping procedure modifies the learning rate schedule compared to baselines. This is unavoidable: the baselines use cosine decay adapted to each training duration (decaying to the same final LR whether training runs for 256 or 2048 steps). **Any early stopping approach would modify the effective schedule**, since stopping while LR remains high prevents complete decay. Our adaptive decay procedure is in fact more faithful to the baseline scheduler, as it ensures both warm-up and decay phases complete within each training run.
> > * **a.1)** To address concerns about confounding factors in the vision setting (where scheduler complexity makes isolation difficult), we conducted additional early stopping experiments in the NLP setting. Our T5 models use a constant LR, enabling straightforward early stopping: we evaluate on a held-out validation set every `num_steps_eval` steps, save the model when validation accuracy reaches a new high, and stop if accuracy fails to improve for `num_evals_patience` evaluations. We tested two configurations and report results in Section 5. **The conclusions hold: early stopping yields significantly better merged models (particularly for TA and TIES) even when expert accuracies are comparable or lower.** In fact, the early-stopped models achieve even larger merging gains than in the vision setting.
> > * **a.2)** Regarding expert accuracy results for early-stopped models: Tables 1 (vision) and 3 (NLP) now include these. Early-stopped experts achieve comparable or lower accuracy than longer-trained experts, yet merging results, especially for Task Arithmetic, TIES, and DARE, are consistently superior for early-stopped experts. This directly demonstrates that early stopping prevents harmful overtraining rather than simply reducing training.
> > * **b)** Early stopping is not intended to guarantee that accuracy will never increase after training halts. It functions as a regularization mechanism that detects when the rate of improvement falls below a meaningful threshold, at which point additional training yields diminishing returns. Even with adaptive LR decay, validation accuracy may continue increasing slowly, but the patience criterion halts training once progress becomes negligible. Early stopping is effective *precisely because* it limits the duration of the late-stage memorization regime, even if marginal accuracy gains remain possible. Our experiments confirm this: early-stopped experts typically achieve comparable or lower accuracy than longer-trained experts, yet consistently yield better merging performance.
> >
> > We thank the reviewer for their feedback. We have revised the manuscript to make our explanations more explicit and to include additional experiments that reinforce our claims. The revisions fully address all raised concerns, and we welcome any further questions.
> >
> > ### References
> > [1] Mansheej Paul, Surya Ganguli, and Gintare Karolina Dziugaite. Deep learning on a data diet: Finding important examples early in training. In M. Ranzato, A. Beygelzimer, Y. Dauphin, P.S. Liang, and J. Wortman Vaughan (eds.), Advances in Neural Information Processing Systems, volume 34, pp. 20596–20607. Curran Associates, Inc.,2021
> >
> > [2] Ben Sorscher, Robert Geirhos, Shashank Shekhar, Surya Ganguli, and Ari Morcos. Beyond neural scaling laws: beating power law scaling via data pruning. In S. Koyejo, S. Mohamed, A. Agarwal, D. Belgrave, K. Cho, and A. Oh (eds.), Advances in Neural Information Processing Systems, volume 35, pp. 19523–19536. Curran Associates, Inc., 2022
> >
> > [3] Idan Attias, Gintare Karolina Dziugaite, Mahdi Haghifam, Roi Livni, and Daniel M Roy. Informa-
> > tion complexity of stochastic convex optimization: Applications to generalization and memoriza-
> > tion. arXiv preprint arXiv:2402.09327, 2024.
> >
> > [4] Vitaly Feldman. Does learning require memorization? a short tale about a long tail. In Proceedings
> > of the 52nd Annual ACM SIGACT Symposium on Theory of Computing, pp. 954–959, 2020.
> >
> > [5] Vitaly Feldman and Chiyuan Zhang. What neural networks memorize and why: Discovering the
> > long tail via influence estimation. In H. Larochelle, M. Ranzato, R. Hadsell, M.F. Balcan, and
> > H. Lin (eds.), Advances in Neural Information Processing Systems, volume 33, pp. 2881–2891.
> > Curran Associates, Inc., 2020. URL https://proceedings.neurips.cc/paper_files/paper/2020/file/1e14bfe2714193e7af5abc64ecbd6b46-Paper.pdf.

---

### Author Response · Authors · 2025-11-26
**Summary of Revisions**

## Summary of Revisions

We have updated our manuscript to incorporate reviewer feedback. Below we summarize the new results and main changes.

### New results
- **Section 5:** Early stopping experiments in the NLP setting
- **Appendix D:** Per-task breakdown of expert and merging accuracies for both vision and NLP settings
- **Appendix E:** New results with additional model architectures (ViT-L-14 for vision and BERT for NLP)
- **Appendix H:** Memorization analysis in early vs. late checkpoints
- **Appendix I:** Quantification of parameter interference throughout training

### Changes to the manuscript
- **Abstract and Introduction:** Clarified contributions
- **Sections 2.1 and 2.2:** Streamlined presentation of background material
- **Section 2.4:** Reordered text to introduce EL2N scores first
- **Section 4:** Rewritten to clarify explanations and integrate new results
- **Section 5:** Added NLP early-stopping results and analysis
- **Section 6:** Moved discussion of model merging and MoErging works to Sections 2.1 and 2.2; added discussion of works modifying the fine-tuning procedure
- **Conclusion:** Clarified contributions and actionable insights
- **Appendix J:** Added pseudocode for the ViT early stopping strategy

We also propose changing the title to "From Memorization to Parameter Interference: How Undertraining Experts Improves Model Merging and MoErging" to better reflect our mechanistic explanation and avoid conflation with unrelated "Less is More" papers.

---

### Author Response · Authors · 2025-12-01

Dear AC,

First, we would like to express our regret regarding the recent incident and the additional burden placed on ACs this year. Given the modified review process, we summarize below how our revisions directly address the initial reviews and why we believe the paper merits acceptance.

**Core Contributions**

Our work provides three central contributions:

1. **Systematic demonstration** that overtraining experts degrades model upcycling (merging and MoErging) across domains (vision & NLP), architectures (ViT-B/L, T5, BERT), model sizes (100M to 3B), and fine-tuning regimes (FFT & LoRA).

2. **Mechanistic explanation** showing that late-stage training is dominated by hard-example memorization, which induces idiosyncratic, task-specific parameter updates that increase parameter interference across experts.

3. **Practical mitigation:** we propose early stopping as a general principle to avoid the high-memorization regime, validated with multiple strategies including constant-LR setups where scheduler confounds are removed.

**Motivation and Significance**

Model merging and MoErging have exploded in popularity, with hundreds of papers over the last 2–3 years. Yet nearly all prior work focuses on designing new merging algorithms, while almost none analyze how fine-tuning itself affects mergeability, even though most open-source models are produced via standard fine-tuning.

Our findings show that training longer, despite improving single-task accuracy, systematically harms upcycling, overturning a widely assumed but untested default. This fills an important gap and provides a new axis for improving upcycling that is orthogonal to method design.

**Potential Impact**

A key outcome is that **choosing the right checkpoint often matters more than choosing the right merging method**. Merging undertrained experts can outperform advanced algorithms applied to overtrained ones (line ~290). This implies:

- A significant share of "merging difficulty" originates upstream in the experts, not downstream in the method.
- Releasing a single additional checkpoint (saved when validation accuracy plateaus) can meaningfully improve downstream upcycling with zero algorithmic overhead—an actionable, low-cost change to current practices.

**Addressing Main Reviewer Concerns**

**1. Mechanism behind why overtraining hurts merging (Reviewers LnVv & HU9q)**

We added substantial new analyses:
- **Appendix H**: Quantifies how late-training updates overwhelmingly target high-difficulty examples, with changes in margin/loss/distribution orders of magnitude larger for hard examples.
- **Appendix I**: Shows longer training systematically increases parameter interference (sign conflicts, overlap, variance).
- **Section 4 rewritten** to explicitly connect: hard-example memorization → idiosyncratic updates → increased parameter interference → worse merging → forgetting of the same hard examples.

Our explanation is also *predictive*: Figure 4 correctly identifies which examples will be forgotten after merging (the hardest ones).

**2. Early stopping is "not novel" (Reviewers LnVv & HU9q)**

We explicitly clarified that:
- We do not claim early stopping as a novel method.
- The contribution is showing *why* early stopping is effective for upcycling—it halts training before the high-interference regime.
- New constant-LR NLP experiments (Section 5) cleanly isolate early stopping from scheduler confounds, yielding significantly improved merging despite equal or lower expert accuracy.

**3. Relation to prior work (Reviewer HU9q)**

HU9q referenced three categories of related work; we clarified how ours differs:
- *Easy-then-hard learning* [E]: Prior works establish data learning order but do not link difficulty to parameter interference or upcycling. We explicitly bridge these literatures.
- *Modified fine-tuning methods* (e.g., SAFT, attention-only) [A, B, C, D]: These works propose new fine-tuning procedures for merging; we analyze the standard procedure used throughout the ecosystem.
- *Works noting long training may hurt merging* [F, G]: These briefly mention the possibility of longer training affecting upcycling but do not study it systematically across domains, architectures, FFT/LoRA, merging/MoErging, nor provide mechanistic analysis.

Our work goes well beyond noting that longer training might affect merging, we establish the generality of this phenomenon across upcycling methods (merging and MoErging), domains (vision & NLP), architectures (ViT-B/L, T5, BERT), model sizes (100M to 3B), and fine-tuning regimes (FFT & LoRA), while providing the mechanistic explanation linking data difficulty and memorization to parameter interference.

---

> ### Author Response · Authors · 2025-12-01
>
> **4. Consensus across positive-reviewers (Reviewers qBGP & AM5C)**
>
> Both positive reviewers rated the paper 6 and praised its clarity, breadth, novel perspective, and practical utility. All remaining requests were fully addressed:
> - Per-task breakdowns (Appendix D)
> - Additional architectures: ViT-L, BERT (Appendix E)
> - Pseudocode for early stopping (Appendix J)
> - Reorganized Sections 2/2.4 and 6 for clarity
>
> **Final Note**
>
> The revised submission resolves all original concerns, significantly strengthens empirical and mechanistic support, and offers practical, impactful insights for the upcycling community. We respectfully ask the AC to evaluate the paper based on the revised version and the extent to which our response addresses the initial review feedback.
>
> We thank all reviewers for their engagement and the AC for their consideration.
>
> **References:**
>
> [A] Jin et al., Fine-Tuning Attention Modules Only: Enhancing Weight Disentanglement in Task Arithmetic. ICLR, 2025.
>
> [B] Lee et al., Mitigating Parameter Interference in Model Merging via Sharpness-Aware Fine-tuning. ICLR, 2025.
>
> [C] Tang et al., Parameter Efficient Multi-task Model Fusion with Partial Linearization. ICLR, 2024.
>
> [D] Ortiz-Jimenez et al.,Task Arithmetic in the Tangent Space: Improved Editing of Pre-Trained Models. NeurIPS, 2023.
>
> [E] Baldock et al., Deep Learning Through the Lens of Example Difficulty. NeurIPS, 2021.
>
> [F] Jyothish Pari, Samy Jelassi, and Pulkit Agrawal. Collective model intelligence requires compatible specialization, 2024.
>
> [G] Luca Zhou, Daniele Solombrino, Donato Crisostomi, Maria Sofia Bucarelli, Fabrizio Silvestri, and Emanuele Rodola. Atm: Improving model merging by alternating tuning and merging, 2025.

---

### Meta-Review · Area_Chair_DbAH · 2026-01-05

**Summary:**

This paper challenges the assumption that improvements at one stage of model merging necessarily propagate to downstream stages. To this end, the authors study the transfer between expert training and upcycling, focusing on broader capabilities and knowledge transfer.

Reviewers raise several concerns. In particular, the explanation for why overtraining degrades merging performance is unconvincing, and the conclusion that “hard samples are learned later” relies on circular reasoning. Additionally, the proposed strategies, early stopping and reducing the learning rate at plateaus, lack novelty, the discussion of related work is insufficient, and the claims are not supported by adequate experimental analysis or theoretical justification.

While the authors addressed some reviewer comments in the rebuttal, the AC finds that the explanation for overtraining remains unconvincing and that substantially more experimental and theoretical analysis is required to support the claims and conclusions. The AC also agrees with the reviewers that the proposed early stopping and learning-rate reduction strategies lack novelty.

Considering these limitations, the AC recommends **rejection** of this paper.

**Reviewer Concerns:**

Reviewers raise several concerns. The explanation for why overtraining degrades merging performance is unconvincing, and the claim that “hard samples are learned later” relies on circular reasoning. Moreover, the proposed strategies of early stopping and learning rate reduction at plateaus lack novelty, the discussion of related work is insufficient, and the claims are not supported by adequate experimental or theoretical analysis.

While the authors addressed some points in the rebuttal, the AC finds that the explanation of overtraining effects remains unconvincing and that substantially more experimental and theoretical evidence is needed to support the conclusions. The AC also agrees that the proposed optimization strategies offer limited novelty.

**Reviewer Scores:**

Given that important concerns remain unresolved, the reviewers are expected to retain their scores after the rebuttal: 2, 2, 6, and 6.

---

### Decision · Program_Chairs · 2026-01-26

Reject